# A Data-Centric Perspective on Evaluating Machine Learning Models for Tabular Data

**Andrej Tschalzev**[*]
University of Mannheim

**Sascha Marton**
University of Mannheim

**Stefan Lüdtke**
University of Rostock

**Christian Bartelt**
University of Mannheim

**Heiner Stuckenschmidt**
University of Mannheim

## Abstract

Tabular data is prevalent in real-world machine learning applications, and new models for supervised learning of tabular data are frequently proposed. Comparative studies assessing the performance of models typically consist of model-centric evaluation setups with overly standardized data preprocessing. This paper demonstrates that such model-centric evaluations are biased, as real-world modeling pipelines often require dataset-specific preprocessing, which includes feature engineering. Therefore, we propose a data-centric evaluation framework. We select 10 relevant datasets from Kaggle competitions and implement expert-level preprocessing pipelines for each dataset. We conduct experiments with different preprocessing pipelines and hyperparameter optimization (HPO) regimes to quantify the impact of model selection, HPO, feature engineering, and test-time adaptation. Our main findings are: **1.** After dataset-specific feature engineering, model rankings change considerably, performance differences decrease, and the importance of model selection reduces. **2.** Recent models, despite their measurable progress, still significantly benefit from manual feature engineering. This holds true for both tree-based models and neural networks. **3.** While tabular data is typically considered static, samples are often collected over time, and adapting to distribution shifts can be important even in supposedly static data. These insights suggest that research efforts should be directed toward a data-centric perspective, acknowledging that tabular data requires feature engineering and often exhibits temporal characteristics. Our framework is available under: https://github.com/atschalz/dc_tabeval.

## 1 Introduction

Since ancient times, tables have been used as a data structure, i.e., to record astronomical observations [69] or financial transactions [6]. Many traditional machine learning (ML) methods, like logistic regression or the first artificial neural networks, were initially developed for tabular data [14, 54, 62]. Even nowadays, in the age of AI, tabular data is the most prevalent modality in real-world applications, including medicine [34], finance [8], manufacturing [75], retail [50], and many others [64, 5]. Several novel deep learning architectures have been contributed in recent years to improve supervised machine learning for tabular data [59, 79, 53, 33, 2, 7, 25, 66, 11, 42, 24].

To evaluate existing approaches, various comparative studies were conducted in recent years [4, 22, 21, 25, 64, 5, 55]. While motivated by different goals, they all have one in common: The focus is on evaluating models on tabular datasets using predefined cross-validation splits and one standardized preprocessing for all datasets. In this paper, we challenge such model-centric evaluation setups by

---

[*]Correspondence to: andrej.tschalzev@uni-mannheim.de

38th Conference on Neural Information Processing Systems (NeurIPS 2024) Track on Datasets and Benchmarks.

highlighting two major limitations (Section 2): 1) The evaluation setups are overly standardized and do not reflect the actual routine of practitioners, which typically includes dataset-specific feature engineering [68]. 2) There is no external reference for the highest possible performance on a task beyond a study's own reporting, which limits its reliability.

To address these issues, we advocate for shifting the research perspective in the tabular data field from model-centric to data-centric. Therefore, our main contribution is an **evaluation framework that includes a collection of ten relevant real-world datasets, dataset-specific expert-level preprocessing pipelines, and an external measure of top performance for each dataset** (Section 3). The datasets were carefully selected by screening Kaggle competitions involving tabular data, and, to our knowledge, our contribution represents the largest existing collection of implemented expert-level solutions for tabular datasets. To assess the potential bias from the first limitation, we **investigate how the model comparison changes when considering dataset-specific preprocessing instead of standardized evaluation setups** (Subsection 4.1). To address the second limitation, we use the leaderboard from Kaggle competitions as an external performance reference and **reassess what is possible with modern methods that were not available when the Kaggle competitions took place** (Subsection 4.2). We find that when considering dataset-specific expert preprocessing, performance differences between the best models shrink, and the importance of selecting the 'right' model diminishes. In addition, we dissect expert solutions for tabular data competitions and **quantify the importance of different modeling components** (Subsection 4.3). We find that measurable progress has been made in automating human effort, but feature engineering is still the most important aspect of many tabular data problems. No model fully automates this aspect and comparisons that don't consider feature engineering merely scratch the surface of the potential performance achievable on many datasets. This paper focuses on independent and identically distributed (i.i.d.) tabular data in line with related work. However, our analysis of Kaggle competitions shows strong evidence that this focus in the research community might not align with practitioners' needs. In particular, we find that **many tabular data competitions on Kaggle have temporal characteristics** (i.e., timestamp features) and we identify test-time adaptation (TTA) as an overlooked but important part of some supposedly static competitions (Subsection 4.4).

Our findings indicate that current academic evaluation setups and benchmarks for tabular data are biased due to their overly model-centric focus. We conclude by **discussing possible directions to improve machine learning for tabular data from a data-centric perspective** (Section 5).

## 2 Related Work

**Machine Learning for tabular data.** Unlike domains like computer vision and natural language processing, an established state-of-the-art neural network architecture does not exist for tabular data [64, 5]. Therefore, recent research has primarily concentrated on developing general-purpose deep learning models often inspired by architectures from other domains [59, 33, 46, 37, 2, 42, 79, 66, 28, 9, 70, 67, 77, 30, 56, 48, 72, 10, 11, 53, 24]. Despite these efforts, Gradient Boosted Decision Trees (GBDTs) remain the state-of-the-art, outperforming even the novel neural models in many studies [5, 26, 55]. This paper aims to motivate more research inspired by tabular data-specific techniques like feature engineering instead of architectures established in other domains.

**Limitations of current evaluation frameworks.** Several benchmarks exist for evaluating tabular machine learning models, focusing on general model comparisons [4, 22, 21, 26, 55] and specific sub-problems [40, 20, 13, 63, 18]. However, these benchmarks do not provide preprocessing settings for the included datasets. Consequently, **most studies adopt a fixed, standardized preprocessing for all datasets** to concentrate on model comparisons [64, 25, 55, 26, 41]. While this model-centric approach is suitable for AutoML, it limits the real-world transferability of model comparisons, as models in practical applications typically follow dataset-specific preprocessing pipelines containing feature engineering techniques [68, 74, 31]. Our evaluation framework is the first to explicitly incorporate a more detailed distinction through diverse preprocessing pipelines. Furthermore, **existing benchmarks lack an external reference (i.e., a leaderboard) for the current best task performance**, hindering comparability across different studies. In contrast, we leverage datasets from ML competitions as an external benchmark for high performance on tasks. Many existing evaluation frameworks **prioritize usability at the expense of representativeness** by limiting sample sizes and removing high-cardinality categorical features, thus evaluating models on artificially constrained dataset versions [4, 26]. Our evaluation framework solely consists of tasks meaningful to the real world without

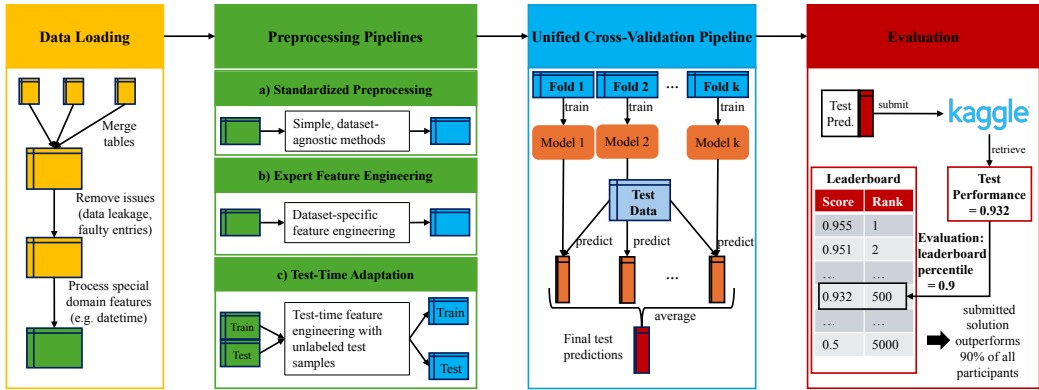

Figure 1: Illustration of the components of our evaluation framework.

imposing artificial restrictions on datasets. Finally, **most evaluation frameworks concentrate on tasks where samples are identically and independently distributed (i.i.d.)**. However, distribution shifts are prevalent in many machine learning applications [45, 71, 73, 51, 78, 20], and adapting to these shifts in tabular data has received limited attention [39, 20]. In this paper, we point out that excluding tabular data with temporal characteristics undermines the reliability of benchmarks, as many real-world applications using the benchmarked models include such data.

**Using Kaggle for model evaluation.** Kaggle is an online platform renowned for its machine learning competitions, hosted by companies and organizations to solve real-world problems in various domains. Some studies have retrospectively compared the performance of new approaches in Kaggle competitions [16, 61, 74]. However, most of these studies are limited to a few competitions or only compared against the leaderboard without investing the high effort of implementing expert solutions. In Subsection 3.1, we will explain that using Kaggle competitions to evaluate new approaches has several benefits. The evaluation framework most similar to ours is presented by Erickson et al. [16], where the proposed AutoML framework was compared to the leaderboard in Kaggle competitions. However, the methods leading to high performance on the leaderboard remain a black box. As we will show, some methods (i.e., test-time adaptation) prevent a fair comparison, and simply evaluating against the leaderboard is not helpful for gaining deeper insights. In contrast, we implement high-performing expert-level solutions, allowing us to dissect the components of interest and truly understand what drives high performance on specific tasks.

## 3 A Data-Centric Evaluation Framework for Tabular Machine Learning

We propose an evaluation framework built upon three crucial aspects that are often overlooked in tabular data research: 1) Evaluation on realistic datasets without removing frequently occurring challenging aspects like high cardinality categorical features, 2) Dataset-specific expert preprocessing pipelines containing feature engineering techniques, and 3) Evaluation against human expert performance on hidden test sets. Figure 1 depicts an overview of our framework. Our design choices are additionally justified by the fact that for each dataset, at least one model in our evaluation ranks among the top 1% of all competition participants (note that not all participants are experts, and leaderboard distributions can vary across datasets).

### 3.1 Collection of Relevant and Challenging Datasets

We rely on the Kaggle community and competitions hosted by companies and institutions to select datasets with expert solutions. Figure 2 illustrates our dataset selection process, and Table 1 summarizes the main properties of the included datasets. Using data from Kaggle competitions has various benefits: 1) The selected tasks are challenging and meaningful to the real world, as companies and institutions only spend money on hosting competitions from which they benefit. 2) Each competition has a clear evaluation setup, including metrics selected to reflect the practitioners' needs. 3) Each competition has a large hidden test set, which has been shown to reduce the risk of adaptive overfitting [61]. 4) The competition leaderboard serves as an external reference for truly high performance,

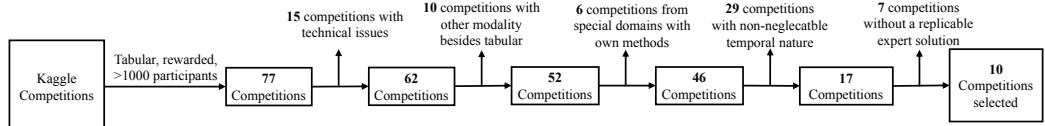

Figure 2: Illustration of the dataset selection process. Details on the criteria and all screened datasets can be found in the Appendix. The Figure only lists the competitions as temporal, which were not already excluded for other reasons. In total, we identified 46 competition datasets with temporal characteristics (i.e., timestamps as a feature). Consistent with related work, we include competitions that have timestamps but can be approached without time-sensitive feature engineering.

| Name | Year | N (train/test) | D (raw/fe) | Categorical | Task | Metric | Model | TTA |
|------|------|----------------|------------|-------------|------|--------|-------|-----|
| MBGM | 2017 | 4209 / 4209 | 377 / 59 | 8 / 47 | Reg | r2 | XGBoost | No |
| SVPC | 2018 | 12296 / 49342 | 4992 / 1420 | 0 / 0 | Reg | rmsle | LGBM | No |
| AEAC | 2013 | 32769 / 58921 | 9 / 315 | 9 / 7,518 | Bin | auc | Ensemble | Yes |
| OGPCC | 2015 | 61878 / 144368 | 93 / 104 | 0 / 0 | Multi | logloss | Ensemble | Yes |
| SCS | 2016 | 76020 / 75818 | 370 / 224 | 0 / 0 | Bin | auc | Ensemble | Yes |
| BPCCM | 2016 | 114321 / 114393 | 132 / 313 | 19 / 18,210 | Bin | logloss | XGBoost | No |
| SCTP | 2019 | 200000 / 200000 | 200 / 600 | 0 / 0 | Reg | auc | NN | Yes |
| HQC | 2015 | 260753 / 173836 | 299 / 300 | 29 / 868 | Bin | auc | XGBoost | No |
| IFD | 2019 | 590540 / 506691 | 432 / 263 | 49 / 13,553 | Bin | auc | CatBoost | Yes |
| PSSDP | 2017 | 595212 / 892816 | 57 / 53 | 8 / 104 | Bin | gini | NN | Yes |

Table 1: Datasets included in our framework. *N* denotes the sample size in thousands, and *D* dimensionality of the raw data and after expert feature engineering. *Categorical* lists the no. of categorical features and the no. of clusters of the highest-cardinality categorical feature. *Metric* corresponds to the competition metric. MBGM and SVPC are regression tasks, OGPCC is a multi-class classification task, and the remaining are binary classification tasks. *Model* corresponds to the best single model used in the original expert solution. For some datasets, no best single model could be distinguished due to the heavy ensembling used. *TTA* denotes if test-time feature engineering has been used in the implemented expert solution.

as many expert teams participated in the competitions. Furthermore, our framework ensures a fair comparison by including a data loading function for each dataset that removes potential side issues, like data leakage or faulty data. This distinguishes our framework from related work that compares approaches to Kaggle solutions [16, 74]. An important insight from screening the competitions is that most tabular datasets had temporal characteristics – i.e., datasets with weak temporal correlations that benefit from time-sensitive feature engineering but not from models with temporal inductive biases (i.e., [32]). This finding will be further discussed in Subsection 4.4.

### 3.2 Expert Solutions and Preprocessing Pipelines

In the context of our paper, *preprocessing* refers to a pipeline that combines a "set of techniques used prior to the application of a [model]" [19]. Feature engineering (FE) refers to techniques that "construct novel features from given data with the goal of improving predictive learning performance" [38]. Consequently, feature engineering is a subset of preprocessing. Our proposed evaluation framework includes three preprocessing pipelines. One is dataset-agnostic and closely resembles the pipelines researchers currently use for model evaluation. The other two are dataset-specific and directly derived from expert solutions. All preprocessing pipelines are model-agnostic, and model-specific preprocessing steps are considered part of the model in our framework.

**Standardized Preprocessing**   The main purpose of this pipeline in our framework is to evaluate single models in a scenario with minimal dataset-specific human effort invested. Continuous missing values are replaced with the mean, and missing categorical feature values are treated as a new category. Furthermore, constant columns are removed, and heavy-tailed targets are log-transformed for regression tasks. As these preprocessing steps are almost universally applied across related work [25, 26, 55], this pipeline represents current evaluation setups in academia well.

**Expert Feature Engineering**    We select one high-performance expert solution from Kaggle for each dataset. The solution was chosen based on the private leaderboard rank and the descriptions' quality and sufficiency. For each solution, we separate the data preparation from the remaining parts of the solution. For most datasets, this pipeline solely consists of feature engineering techniques. Besides a few distinctions between tree-based and deep learning models, the pipelines are model-agnostic. Model-specific preprocessing steps (i.e., feature normalization for neural networks) are considered part of the model in our framework and are explained in the Appendix. This paper focuses on a pipeline perspective and does not discuss single feature engineering steps further. Implementation details and feature engineering techniques used for specific datasets are provided in the Appendix and in our publicly available code. For this pipeline, we ensured that all feature engineering operations included were on the training data and that a model could have learned the same patterns without external information. Some of the feature engineering techniques used by the experts occur across multiple datasets: groupby interactions of categorical and numeric features (4), 2-order categorical interactions (3), feature selection (3), categorical frequency encoding (3), dimensionality reduction (2), 3-order categorical interaction (2), 2-order arithmetic interactions (2), sum of missing values in a row (2), and sum of zeros in a row (2). A common pattern is that the most frequently applied feature engineering steps include categorical features and that feature interactions are frequently manually engineered while transformations of single features are rare.

**Test-Time Adaptation**    This pipeline is exactly the same as the expert feature engineering pipeline, with the key difference that the test data is used for feature engineering where applicable. Most ML competitions are organized so that the test features (but not the targets) are given. We found that the top solutions used the test data in their data preparation for six of the datasets in our framework. Hence, this pipeline represents the actual preprocessing used by the experts. While this might be considered an unfair and unrealistic setup, there are applications where using unlabeled test data for unsupervised learning is applicable (see Appendix A.3). We argue that this conceptualization makes many tabular ML competitions a test-time adaptation (TTA) task. TTA is a type of domain adaptation where test samples are used at test time in an unsupervised or self-supervised way to update or retrain a model [76, 44, 43, 65, 57]. We term the common Kaggle practice of engineering domain-invariant features at test time as **test-time feature engineering**. The feature engineering techniques most frequently used to this end are groupby interactions, frequency encodings, and learning joint low-dimensional representations. With this preprocessing pipeline, we are the first to closer examine test-time feature engineering in Kaggle competitions.

### 3.3    Modeling and Evaluation Framework

**Modeling Pipeline and Models**    We implement a unified modeling pipeline for all datasets with a dataset-specific cross-validation (CV) ensembling procedure. The validation sets are used for early stopping and determining the best hyperparameters. The final test data predictions are an ensemble of averaging the test predictions of each fold. We use three gradient-boosted tree libraries (*XGBoost* [12], *LightGBM* [36], and *CatBoost* [60]) because each was used in at least one of the expert solutions. Each expert solution that used neural networks developed a highly customized network for the particular competition. We want to assess whether recently developed general-purpose architectures can replace the high effort of building custom networks. Hence, we chose *ResNet* and *FTTransformer* [25] because they have been frequently used in recent benchmark comparisons and have shown strong performance [26, 55]. Because the Resnet essentially is an MLP with skip connections, it serves as a baseline representing what was already possible before the recent developments in DL for tabular data. In addition, we use two more recent approaches: *MLP-PLR* [23], which can help learn high-frequency functions, mitigating a major weakness of deep learning for tabular data [26]; and *GRANDE* [53], a recent representative of hybrid neural-tree models. Although other recent architectures exist, we don't include more, as our focus is not on benchmarking particular models but rather on demonstrating the importance of data-centric evaluation. To assess how well fully automated solutions perform without any preprocessing, we additionally evaluate *AutoGluon*, which has been shown to be the current best AutoML solution [21].

**Hyperparameter Optimization**    Hyperparameter optimization is done per fold to obtain a diverse CV ensemble. Each model is evaluated in three HPO regimes: 1) *Default*: Either library default or hyperparameters suggested in related work, 2) *Light HPO*: 20 random search iterations. 3) *Extensive HPO*: 20 random search warmup iterations + 80 iterations of the tree-structured Parzen estimator algorithm [1]. More details on the hyperparameter optimization can be seen in the Appendix.

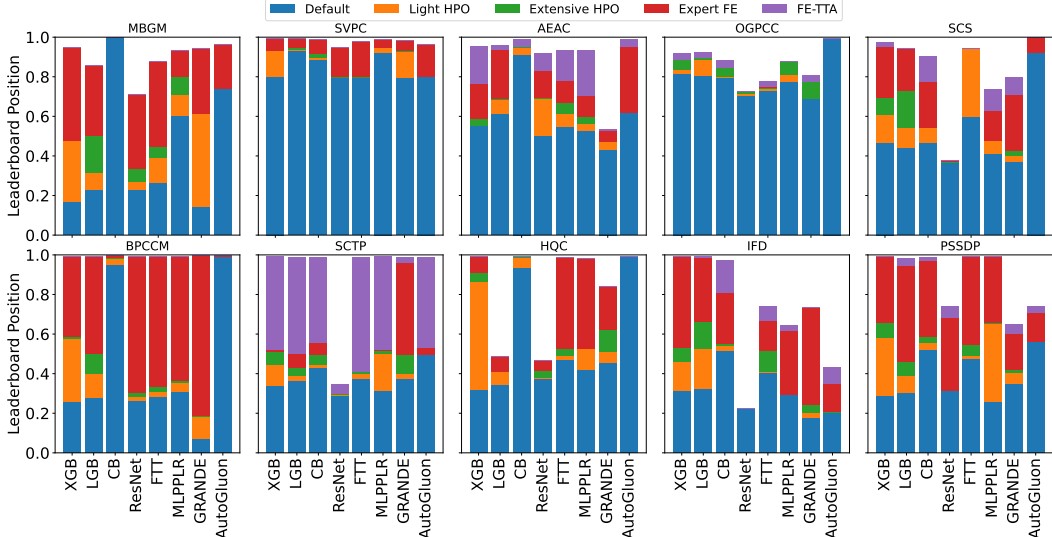

Figure 3: Performance gains from different modeling components on the private Kaggle leaderboard by dataset and model. Higher values correspond to a better position. Each segment represents the performance gain of adding the modeling component to the previous configuration. 'Default' corresponds to the model performance with default hyperparameters in a standardized preprocessing pipeline. Light and extensive HPO correspond to tuning hyperparameters in the same preprocessing pipeline. Expert FE and FE-TTA correspond to the model performance with extensively tuned hyperparameters in the feature engineering and the test-time adaptation pipeline respectively.

**Evaluation**  We use the Kaggle API to automatically submit predictions and retrieve performance results after evaluating against the hidden targets. Each dataset is evaluated on the metric specified by the competition host. Instead of reporting this metric directly, we report the solution's private leaderboard position as the percentile. This has the benefit that although different metrics are used to evaluate the model, comparisons across datasets are possible. Note that the leaderboard position is always a snapshot of the end of each competition. In the Appendix, we additionally report performances on the actual metrics for each dataset. Throughout the paper, higher values represent a better performance (leaderboard position). As we use only one test set per dataset and less datasets compared to academic benchmarks, concerns about overfitting might be raised. However, ten datasets with one test set are less of an issue in our framework than it would be in conventional benchmarks, because: 1) The datasets in our framework, especially the test data, are comparably large and overfitting them is harder. Roelofs et al. [61] found that at least 10,000 test examples is a reasonable minimum test set size to protect against adaptive overfitting in Kaggle challenges. All test sizes in our framework, except for the MBGM dataset, are at least  of size 50,000. 2) Test labels are unknown making it hard to purposefully overfit on particular samples. 3) The need of submitting to Kaggle, although automated, is an additional overfitting barrier.

## 4   Experimental Evaluation

Our framework allows us to assess the dataset-specific individual performance impact of model selection, hyperparameter optimization, feature engineering, and test-time adaptation. Whenever not stated otherwise, we report the results for the extensively tuned hyperparameter setup. As a general overview, Figure 3 shows how each of the analyzed modeling components improves over the default baseline for each model and dataset. The length of each segment indicates the performance gain relative to the previous configuration. The results demonstrate the importance of an external performance reference: If we only considered the standardized evaluation setup (blue/orange/green segments), we would only be scratching the surface of achievable task performance for many data sets.

### 4.1 How Model Comparisons Change When Considering Dataset-specific Preprocessing

Three observations stand out when evaluating models in different preprocessing pipelines (Figure 4). **1) The model rankings change considerably**, as indicated by the relatively low Spearman coefficients between the standardized preprocessing pipeline and the other pipelines. **2) The performance gap between all models diminishes when considering expert preprocessing.** On average, all models benefit from feature engineering, and multiple models can reach top performance. While all models benefit from TTA, the performance increase varies. **3) The superiority of CatBoost vanishes when considering dataset-specific preprocessing.** The reason is that CatBoost already incorporates specific feature engineering steps in its algorithm for which other models need manual engineering, as we will further elaborate in Subsection 4.3.

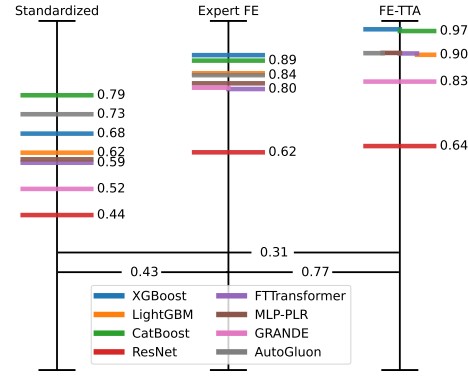

Figure 4: Average leaderboard position of models with different preprocessing. Black horizontal lines denote the Spearman correlation between all results with the respective preprocessing.

### 4.2 Measurable Progress Through Recent Efforts

Figure 5 shows the model ranking on the private Kaggle leaderboard when trained after standardized preprocessing. CatBoost achieves top ranks in three competitions (MBGM, BPCCM, HQC) where a high manual effort in feature engineering was previously necessary. Similar to Erickson et al. [16], AutoGluon achieves top ranks in two of these (BPCCM, HQC) and one additional competition (OGPCC). Regarding neural networks, novel architectures rank higher than the ResNet baseline on nine datasets. In two competitions, neural networks were originally the single-best models (SCTP, PSSDP - see Table 1). In our analysis, MLP-PLR and FTTransformer are able to reach top ranks on these datasets after feature engineering and test-time adaptation, while ResNet performs worse (see Figure 3). All neural networks originally used in the competitions were custom-designed for the particular competition. Hence, our analysis confirms that meaningful progress has been made in developing general-purpose architectures for tabular data as they reduce the necessity of custom-designed networks. Although the progress in the tabular data field is clearly visible, top performance cannot be reached without human effort for six datasets.

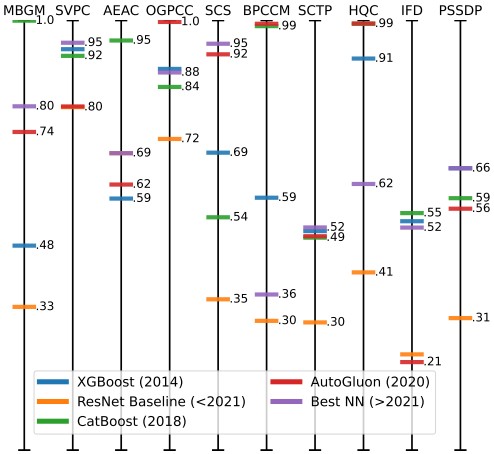

Figure 5: Progress made through recent models trained in the standardized preprocessing pipeline, illustrated by retrospective comparison to the Kaggle leaderboard. Best NN denotes the best model of FTTransformer, MLP-PLR, and GRANDE.

### 4.3 Feature Engineering is Still the Most Important Factor for Top Performance

**The most remarkable performance gains are achieved through feature engineering.** Figure 6 shows that expert feature engineering is the most important modeling component on average. This holds true for all models, indicating that unlike for modalities like imaging, neural networks do not automate feature engineering for tabular data. When comparing the performance of different models in the standardized preprocessing pipeline (blue/orange/green bars in Figure 3), we can observe that using any other model than CatBoost rarely brings large gains. Only for the SCS dataset does FTTransformer clearly outperform all other models. For all other datasets, the average performance gains achievable solely with model selection are small.

Hence, our results confirm the findings of McElfresh et al. [55] that model selection is less important than HPO on a strong tree-based baseline for most datasets. Furthermore, we extend this finding by quantifying the even more important aspect of dataset-specific feature engineering.

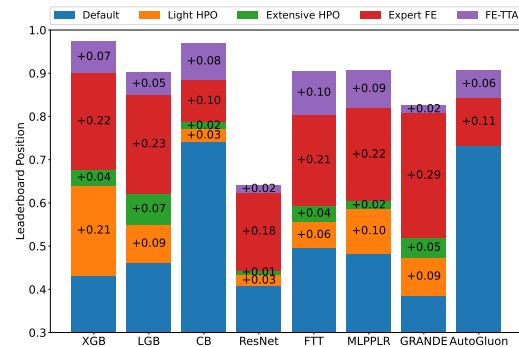

Figure 6: Leaderboard performance gains from different modeling components per model. 'Default' corresponds to the model with default hyperparameters. The results for Expert FE and FE-TTA are reported after extensively tuning hyperparameters.

**Feature engineering is responsible for the high performance of CatBoost.** Our analysis of different preprocessing pipelines reveals that CatBoost benefits much less from feature engineering than other models. The reason is that CatBoost incorporates explicit feature engineering techniques in its learning procedure. In particular, counts and target-based statistics are used to generate encodings for categorical features, and combinatorial encoding methods capture categorical feature interactions [60]. When considering the same feature engineering techniques for the other models, the gap to CatBoost drastically shrinks for most models, and XGBoost performs similarly to CatBoost on average. Hence, CatBoost's success in recent benchmarking studies [55, 20] can, at least to some extent, be attributed to feature engineering.

**The optimal treatment of categorical features can be dataset-specific.** Table 2 shows that a different treatment of categorical features than the model-inherent treatment was necessary for two datasets to achieve top performance. Furthermore, each of the two datasets required a different encoding method. This shows that standardized preprocessing can be biased for categorical features. Furthermore, the performance of deep learning models on categorical data can be greatly improved with feature engineering techniques on categorical data. I.e., for the AEAC dataset, which consists entirely of categorical features, all neural networks gain from feature engineering techniques like categorical feature interactions, as can be seen in Figure 3. This suggests that current architectures do not adequately capture the complex patterns within categorical data. Hence, whenever the goal is not to evaluate models as AutoML solutions, categorical data treatment methods in comparative studies should not only be model-specific, but also dataset-specific.

|  | PSSDP | | BPCCM | |
|---|---|---|---|---|
|  | Default | OHE | Default | Target |
| XGBoost | 0.69 | **0.99** | 0.4 | **0.99** |
| LightGBM | 0.54 | **0.94** | 0.35 | **0.99** |
| CatBoost | 0.71 | **0.97** | **1.0** | **1.0** |

Table 2: Performance of tree-based models with different categorical data treatment methods. 'Default' corresponds to the model-inherent method.

## 4.4 The Importance of Test-Time Adaptation and Temporal Characteristics

**Test-time feature engineering consistently improves the performance of single models.** Table 3 shows that test-time feature engineering leads to performance gains over solely using the train data for feature engineering for all datasets. From the task perspective, the feature engineering used for AEAC and OGPCC only leads to performance gains when used as a test-time adaptation method. This shows that some of the feature engineering techniques used in Kaggle competitions actually serve the purpose of test-time adaptation. For three datasets, ranking among the top 1% on the leaderboard was not achieved without test-time adaptation. Our results indicate that simply comparing approaches to the Kaggle

|  | Best single model | | | AutoGluon | | |
|---|---|---|---|---|---|---|
|  | Stand. | FE | TTA | Stand. | FE | TTA |
| AEAC | 0.953 | 0.937 | 0.991 | 0.618 | 0.953 | **0.993** |
| OGPCC | 0.896 | 0.871 | 0.923 | **0.996** | 0.983 | 0.995 |
| SCS | 0.945 | 0.953 | 0.975 | 0.92 | 0.999 | **1.0** |
| SCTP | 0.518 | 0.962 | **0.992** | 0.498 | 0.531 | 0.991 |
| IFD | 0.662 | 0.988 | **0.992** | 0.205 | 0.351 | 0.432 |
| PSSDP | 0.656 | 0.994 | **0.995** | 0.562 | 0.707 | 0.742 |

Table 3: Performance comparison in different preprocessing pipelines with a focus on top performance. AutoGluon is displayed separately to prevent bias in the single-model comparison.

leaderboard, as done in previous studies [16, 74], is insufficient. Techniques like test-time adaptation are frequently used in Kaggle competitions and limit comparability to approaches that don't use the test data. Hence, a fair model comparison to expert solutions using the Kaggle leaderboard can only be ensured under controlled conditions through implemented expert solutions such as our pipelines.

**Models in real-world applications are often applied to non-i.i.d. tabular data.** By definition, TTA should only be effective if the data violates the i.i.d. assumption and contains distribution shifts to adapt to. Indeed, the data collection process likely happened over time for most of the datasets used in our framework. However, timestamps were not always provided as the competitions were conceptualized as static tabular data tasks. Therefore, most of the datasets were also used in at least one comparative study for tabular data, although non-i.i.d. was a criterion for exclusion (e.g., SVPC, AEAC, and PSSDP in [21], SCTP and OGPCC in [41], or MBGM in [26]). Our results show, that despite treating datasets as static, the samples remain non-i.i.d. and approaches like test-time adaptation can improve performance. Furthermore, there is evidence that other datasets treated as i.i.d. in related work actually have a temporal nature. I.e., the electricity dataset [29] is frequently used in academic benchmarks [26, 55]. At the same time, this dataset would actually require a time-based data split and is used as a benchmark in online learning to measure the ability of models to adapt to concept drifts [15]. As models for tabular data assume the data to be i.i.d., most benchmarks for evaluating tabular general-purpose models either directly name the data being non-i.i.d. as an exclusion criterion [26, 21] or exclude data that requires special CV procedures [4], which leads to the same results. In contrast, our analysis of Kaggle competitions revealed that most tabular data competitions have temporal characteristics and that the best solutions for such datasets typically engineer time-invariant features and utilize tabular data models assuming the data to be i.i.d (i.e. [32]). We conclude that there might be a mismatch between current evaluation frameworks for tabular data in academia and the tabular data tasks practitioners were interested in getting solved through ML competitions on Kaggle.

## 4.5  Limitations

Despite the outlined advantages our evaluation has some limitations compared to evaluation designs in academic benchmarks:

1. The distribution of leaderboard rankings varies across competitions, and advancements in one competition's leaderboard do not necessarily translate equivalently in another. Moreover, large gains on a leaderboard may correspond to only minor increases on the actual task metric, particularly in competitions where solutions are widely shared and numerous similar entries are submitted. As an alternative, we repeat our evaluation based on the original task metric in Appendix F.

2. Not all leaderboard submissions are made by experts, resulting in a tail of lower-quality entries in each competition. Users of our framework should exercise caution when interpreting the leaderboard as a performance metric, ensuring that only factually accurate statements about its implications are used.

3. Due to the extent of our experiments, it was infeasible to repeat them to obtain error bars. This limitation is important, as it leaves the randomness introduced by data splits, model-specific characteristics, and hyperparameter optimization unquantified. Hence, small differences between models on single datasets need to be interpreted with caution. Nevertheless, our primary focus was to assess the effects of various modeling components. In this regard, the extensive nature of our experiments and the clear distinctions observed between preprocessing pipelines across multiple models and datasets suggest a minimal impact of randomness on our main findings. For users applying our framework to systematically compare individual models, alternative configurations with repeated evaluations would be necessary to increase reliability.

4. Due to the focus on incentivized competitions, most datasets are from the finance domain and primarily represent North America and Europe, leading to an underrepresentation of other domains and regions. To address this limitation, future analyses could incorporate competitions from additional platforms. However, as our framework was designed to be user-friendly, we focused on Kaggle, which contains an API for effortlessly downloading datasets and submitting predictions.

A more extensive discussion of limitations can be found in Appendix E.

# 5 Implications for Future Work

We challenged the prevalent model-centric evaluation setups in tabular data research by comparing evaluations with standardized preprocessing pipelines to evaluations with expert preprocessing pipelines. We have shown that current research is overly model-centric, while tabular datasets often require dataset-specific feature engineering or violate the i.i.d. assumption the models are based on. This reveals important insights and directions for future work in Machine Learning for tabular data.

**More careful choice of preprocessing for model evaluation.** Our findings highlight that standardized evaluation setups do not necessarily ensure fair model comparisons. In standardized preprocessing setups, models are evaluated as if they were AutoML solutions, whereas in real-world applications, they are components of highly dataset-specific pipelines. Researchers should be aware a) whether their datasets are amenable to feature engineering, b) that standardized preprocessing setups treat models as AutoML systems, and, c) that true ceteris paribus (c.p.) comparisons are hard if some models (i.e., CatBoost) apply feature engineering internally and others don't. Feature-engineered evaluation setups can be more suitable if a study aims for truly c.p. conditions or for evaluating models in realistic scenarios. Standardized evaluation setups are more suitable for benchmarking AutoML solutions and can also be suitable if models are expected to learn features without human effort. Future research could emphasize incorporating dataset-specific (expert) preprocessing pipelines into benchmarks or separate raw data benchmarks from fully preprocessed benchmarks, as done in our study. However, gathering high-quality expert solutions at a large scale is tedious and may require a community effort.

**Need for external performance references.** Our analysis shows that evaluations without considering the highest achievable performance on a task don't actually measure the state-of-the-art. Despite numerous benchmarks, there is no established standard to measure progress. A benchmark with a public leaderboard and a dynamic collection of meaningful and unsolved real-world datasets could facilitate progress.

**Investigate why some feature engineering operations are not inherently learned by models.** Researchers developing general-purpose models should recognize the impact of feature engineering on model performance. CatBoost has advanced the field by automating feature engineering on categorical data. However, significant feature engineering effort is still necessary for datasets where this is not the only challenge. Our study made it evident that there are transformations of the feature space which are not learned by models without manual feature engineering. While we focused on a pipeline perspective, future work could look at particular feature engineering techniques to uncover modes of failure for current models and develop novel architecture components. I.e., our experiments show that deep learning models benefit from feature engineering on categorical features. Hence, unlike previously claimed [26], categorical features can indeed be an important challenge for deep learning models and future work could focus on improvements over conventional embeddings. Our expert feature engineering pipelines can serve as a starting point for evaluating and developing new methods. Furthermore, AutoGluon was sometimes outperformed by single models in our analysis, although it contains the same models. Future work could investigate why AutoGluon does not always benefit from feature engineering to the same extent as single models.

**Methods for tabular data with temporal characteristics.** Our analysis highlights the temporal nature of many real-world tabular data tasks, as well as the importance of accounting for distribution shifts. Future work could investigate test-time adaptation methods specifically for tabular data, using our datasets and the identified test-time feature engineering techniques as baselines. Furthermore, our findings indicate that the current research focus on static i.i.d. data might hinder the development of techniques to handle weak temporal correlations in tabular data. Future work should focus on developing models with inductive biases for tabular data with temporal characteristics.

**Align tabular benchmarks with practitioners needs.** We have shown that models developed for tabular data are often applied to datasets with temporal characteristics, while existing tabular data benchmarks are overly focused on i.i.d. data. General-purpose tabular benchmarks should consider including tabular datasets with temporal characteristics instead of excluding them. Furthermore, a benchmark solely for tabular datasets with temporal characteristics could significantly advance model development for this relevant data problem.

## Acknowledgments and Disclosure of Funding

This research was supported by the Ministry of Economic Affairs, Labour and Tourism Baden-Württemberg. Furthermore, the authors acknowledge support by the state of Baden-Württemberg through bwHPC and the German Research Foundation (DFG) through grant INST 35/1597-1 FUGG. We want to thank the reviewers of the NeurIPS datasets and benchmarks track for their invaluable feedback, which greatly improved the quality of this paper. Moreover, we want to thank Kaggle for hosting the competitions used in our framework and maintaining them as open source. In addition, we thank the Kaggle users gmobaz, joeytaj, pyduan, bensolucky, mikeskim, leustagos, efimov, titericz, davutpolat, jiweiliu, kazanova, stasg7, mmueller, kyazuki, utility, raddar, confirm, psilogram, fl2ooo, konradb, rejulien, cdeotte, kyakovlev, xiaozhouwang, kelexu, and qqgeogor for sharing the competition solutions we used. Lastly, we thank Malte Gaber and Michael Temnov for their assistance in implementing Kaggle solutions.

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

# A  Datasets and Expert Solutions

In this Section, we provide more details on the dataset/competition selection process and the expert solutions implemented in our framework. Table 4 shows the name of all Kaggle competitions along with additional information.

## A.1  Dataset Selection

The main paper already provides an overview of the main characteristics of the selected datasets and our main selection criteria. In this Subsection we further explain the selection criteria and summarize the excluded datasets.

| Name | Competition Name | End date | Expert solution | openml-id |
|------|------------------|----------|-----------------|-----------|
| MBGM | mercedes-benz-greener-manufacturing | 2017-06-11 | 1st place | 42570 |
| SVPC | santander-value-prediction-challenge | 2018-08-21 | 6th place | 42572 |
| AEAC | amazon-employee-access-challenge | 2013-08-01 | 1st place | 4135 |
| OGPCC | otto-group-product-classification-challenge | 2015-05-19 | 8th place | 45548 |
| SCS | santander-customer-satisfaction | 2016-05-03 | 3rd place | - |
| BPCCM | bnp-paribas-cardif-claims-management | 2016-04-19 | 8th place | - |
| SCTP | santander-customer-transaction-prediction | 2019-04-11 | 1st place | 42395 |
| HQC | homesite-quote-conversion | 2016-02-09 | 15th place | - |
| IFD | ieee-fraud-detection | 2019-10-04 | 1st place | - |
| PSSDP | porto-seguro-safe-driver-prediction | 2017-11-30 | 2nd place | 43121 |

Table 4: Datasets and expert solutions included in our framework. The competitions can be accessed at https://www.kaggle.com/competitions/{Competition Name}. The openml id is provided for better contextualization with prior work.

In an initial search through the competitions hosted on Kaggle, we selected all datasets that satisfied the following criteria:

- **Tabular**: We only consider competitions that include tabular data.

- **Popular competitions**: We consider all competitions with at least 1000 participants.

- **Additional incentive**: We only consider competitions that are incentivized, either monetarily or otherwise.

We identified 77 competitions that satisfy these criteria and further applied dataset-specific criteria to select competitions for our framework. Table 5 summarizes all excluded datasets. In the following, we explain the exclusion criteria:

- **Technical Issues**:
  - **Code competition**: Automating solution submission is non-trivial since the competition is a code competition with special requirements. (3 competitions)
  - **Ongoing**: The home-credit-credit-risk-model-stability competition was not finished at the time of the development.
  - **Availability**: Dataset not available anymore (6 competitions)
  - **Sample size**: The restaurant-revenue-prediction dataset had only 137 training samples, preventing reliable non-random model comparisons.
  - **Leak**: The competition was won through an unresolvable data leak s.t. a fair evaluation is impossible. (3 competitions)
  - **Submission error**: Submitting to Kaggle doesn't work due to an unresolved error for the liberty-mutual-group-property-inspection-prediction competition.
- **Other Modality**: Utilization of other modalities, e.g. images, text, signals, molecular, or genetic data, was a major part of the expert solution besides tabular data. (10 competitions)
- **Special domain**:
  - **Spatial**: The data has spatial correlations that cannot easily be learned by the existing general-purpose models. (4 competitions)

- **Recommendation**: Click-through-rate prediction and recommendation tasks were excluded because dedicated models exist for these tasks (e.g., [27, 47, 52]), while we focus on general-purpose models. (avazu-ctr-prediction and kkbox-music-recommendation-challenge)

- **Temporal**: In line with related work, we focus on i.i.d. tabular data. Competitions where time-sensitive feature engineering was the key to competition success were excluded. This also includes competitions without an explicit timestamp where multiple tables needed to be merged, and where the strategy for merging datasets was a relevant part of the solution, e.g., due to specific aggregation strategies. Although no timestamps are available for those datasets, the underlying task necessitating merging was temporal. An example is the elo-merchant-category-recommendation competition. (32 competitions)

- **Expert Solution availability/reproducibility**: These datasets could be included in our framework but, for various reasons, could not be used with different preprocessing pipelines:
  - For the walmart-recruiting-trip-type-classification, no top 1% solution was available.
  - For the Springleaf-marketing-response and the ClaimPredictionChallenge datasets, solution descriptions were available but were insufficient to reproduce the solution.
  - For the prudential-life-insurance-assessment dataset, the main aspect for high performance was calibration and transforming the target to simplify calibration, which was out-of-scope in our framework.
  - For two competitions, the expert solution mainly consisted of heavy ensembling on different dataset versions and models, which we could not reproduce within our setup (allstate-claims-severity, higgs-boson).
  - For the sberbank-russian-housing-market competition, high performance was mainly achieved by training different models for different samples in a dataset and by modifying the target in highly task-specific ways.

| Competition Name | No. Teams | Exclusion criteria |
|---|---|---|
| home-credit-default-risk | 7176 | Temporal |
| icr-identify-age-related-conditions | 6430 | Code competition |
| m5-forecasting-accuracy | 5558 | Temporal |
| amex-default-prediction | 4874 | Temporal |
| LANL-Earthquake-Prediction | 4516 | Other Modality, Temporal |
| optiver-trading-at-the-close | 4436 | Temporal |
| lish-moa | 4373 | Code competition |
| jane-street-market-prediction | 4245 | Availability, Temporal |
| elo-merchant-category-recommendation | 4110 | Temporal |
| talkingdata-adtracking-fraud-detection | 3943 | Temporal |
| optiver-realized-volatility-prediction | 3852 | Temporal |
| zillow-prize-1 | 3770 | Spatial |
| ashrae-energy-prediction | 3614 | Temporal |
| ga-customer-revenue-prediction | 3611 | Temporal |
| godaddy-microbusiness-density-forecasting | 3547 | Temporal |
| petfinder-pawpularity-score | 3537 | Other Modality |
| home-credit-credit-risk-model-stability | 3481 | Ongoing |
| rossmann-store-sales | 3298 | Temporal |
| sberbank-russian-housing-market | 3264 | No expert solution |
| allstate-claims-severity | 3045 | No expert solution |
| h-and-m-personalized-fashion-recommendations | 2952 | Other Modality, Temporal |
| two-sigma-financial-news | 2927 | Availability, Temporal |
| ubiquant-market-prediction | 2893 | Availability, Temporal |
| champs-scalar-coupling | 2737 | Other Modality |
| predict-energy-behavior-of-prosumers | 2731 | Temporal |
| instacart-market-basket-analysis | 2621 | Temporal |
| prudential-life-insurance-assessment | 2610 | No expert solution |
| otto-recommender-system | 2574 | Temporal |
| novozymes-enzyme-stability-prediction | 2482 | Other Modality |

| Competition Name | No. Teams | Exclusion criteria |
|---|---|---|
| two-sigma-connect-rental-listing-inquiries | 2480 | Other Modality, Temporal |
| microsoft-malware-prediction | 2410 | Temporal |
| mercari-price-suggestion-challenge | 2380 | Other Modality |
| predicting-red-hat-business-value | 2260 | Leak, Temporal |
| restaurant-revenue-prediction | 2257 | Sample size |
| liberty-mutual-group-property-inspection-prediction | 2232 | Submission error |
| springleaf-marketing-response | 2221 | No expert solution |
| recruit-restaurant-visitor-forecasting | 2148 | Temporal |
| home-depot-product-search-relevance | 2123 | Leak |
| two-sigma-financial-modeling | 2063 | Availability, Temoral |
| predict-student-performance-from-game-play | 2051 | Temporal |
| jpx-tokyo-stock-exchange-prediction | 2033 | Temporal |
| petfinder-adoption-prediction | 2023 | Other Modality |
| expedia-hotel-recommendations | 1971 | Spatial |
| grupo-bimbo-inventory-demand | 1963 | Temporal |
| g-research-crypto-forecasting | 1946 | Temporal |
| avito-demand-prediction | 1868 | Other Modality, Temporal |
| amp-parkinsons-disease-progression-prediction | 1805 | Code competition, Temporal |
| higgs-boson | 1784 | No expert solution |
| santander-product-recommendation | 1779 | Temporal |
| talkingdata-mobile-user-demographics | 1680 | Leak |
| favorita-grocery-sales-forecasting | 1671 | Temporal |
| avazu-ctr-prediction | 1602 | Recommendation |
| allstate-purchase-prediction-challenge | 1566 | Temporal |
| axa-driver-telematics-analysis | 1524 | Availability, Temporal |
| new-york-city-taxi-fare-prediction | 1483 | Spatial |
| airbnb-recruiting-new-user-bookings | 1458 | Temporal |
| vsb-power-line-fault-detection | 1445 | Temporal |
| bosch-production-line-performance | 1370 | Temporal |
| hhp | 1350 | Availability |
| predict-west-nile-virus | 1304 | Temporal |
| ClaimPredictionChallenge | 1278 | No expert solution |
| nyc-taxi-trip-duration | 1254 | Spatial, Temporal |
| PLAsTiCC-2018 | 1089 | Temporal |
| kkbox-music-recommendation-challenge | 1081 | Recommendation |
| foursquare-location-matching | 1079 | Other Modality |
| coupon-purchase-prediction | 1072 | Temporal |
| walmart-recruiting-trip-type-classification | 1043 | No expert solution |

Table 5: Datasets excluded during the selection process

## A.2 Implemented Components of Expert Solutions

In this Subsection, we document the components of our framework that were directly derived from expert solutions.

**Task conceptualization in data loading.** For all datasets, the data loading includes merging tables, defining the target, and defining categorical features. For some datasets, we incorporated parts of expert solutions into the task conceptualization as a part of the data-loading function:

- mercedes-benz-greener-manufacturing: The index is used as a numeric feature as it was necessary to score top leaderboard ranks.

- santander-value-prediction-challenge: 1) The target is marked as heavy-tailed to be transformed in the standardized preprocessing pipeline. 2) There was a data leak allowing to derive the test targets for some samples. The top expert solutions used these samples for data augmentation. Hence, we also moved these samples from the test to the training dataset, s.t. this leak is not an issue for any of our pipelines.

- homesite-quote-conversion: Extract weekday from datetime feature.

- porto-seguro-safe-driver-prediction: Replace -1 with nan.

**Feature Engineering Pipelines.** For each expert solution, we extract the data preparation, which mostly consisted of feature engineering. The expert solutions of the datasets contained the following feature engineering operations:

- mercedes-benz-greener-manufacturing: Addition of binary features, logical_and of binary features, sum of multiple binary features, feature selection

- santander-value-prediction-challenge: {max, mean, min, median, first nonzero, last nonzero, no. of nans, no. of unique values} of groups of multiple features. The groups mostly consisted either of 40 or 99 features. Three groups were formed with 4991, 991, and 4000 features. The groups were previously determined based on expert knowledge. However, all operations to obtain the new features could theoretically be learned solely from the train data, and no timestamps are given explicitly.

- amazon-employee-access-challenge: (normalized) groupby interactions, 2- and 3-order categorical interactions, (normalized) frequency encoding, frequency encoding of interactions, log of frequency features, drop constant features

- otto-group-product-classification-challenge: tSNE features, PCA features, KMeans centroid features

- santander-customer-satisfaction: a few data cleaning steps, Remove highly correlated and constant features, remove features with <4 target=1 instances, count of value 0/3/6/9 in a row, percentile rank of feature A within feature B (considered a special kind of groupby interaction), ratios, (X mod 3) == 0, KMeans features with 2-11 clusters, binary feature separating population based on different other feature values

- bnp-paribas-cardif-claims-management: 2- and 3-order categorical interactions, Convert numerical to categorical by rounding, 2-order Arithmetic combinations, 11-order categorical interaction, out-of-fold target encoding

- santander-customer-transaction-prediction: replacing values that are unique in train data (added test for test-time adaptation) with the mean of the feature, Extract categorical features from numeric. Features have four (five if test data used) categories: 1) value appears at least another time in data with target==1 and no 0, 2) value appears at least another time in data with target==0 and no 1, 3) value appears at least two more time in data with target==0 & 1, 4) value is unique in data (if test-time adaptation: 5) value is unique in data + test)

- homesite-quote-conversion: sum NAs in a row, sum of zeros in a row, two-order categorical interaction

- ieee-fraud-detection: feature selection, normalize "time deltas" from some point in the past (Feature 1 (F1)-Feature 2 (F2)/(24*60*60)), frequency encoding (train & test), label encode categoricals, groupby interactions (mean, std, count), 2-way categorical interactions, (F1 - floor(F2), F1(cat) + ascat(floor(F2)-F3) - is not used directly, but for more aggregations), abs(F1-F2)>3, use cat features as numeric

- porto-seguro-safe-driver-prediction: Feature selection, sum of missing values, frequency encoding of high-order interaction of categorical features, only for tree-based models: one-hot-encoding of categorical features, only for neural networks: train XGBoost models with one group of features as input and another feature as output - use the out-of-fold predictions as features

For the ieee-fraud-detection competition the winning solution found that once one transaction of a customer is a fraud in the train dataset - all are. They deal with that by implementing a postprocessing function labeling all customers as a fraud whenever one of the transactions is a fraud. As this pattern could also be learned by models, we decided to treat this aspect as part of the expert preprocessing pipeline.

The treatment of categorical features is left to the models whenever possible. The utilized encoding is listed as part of the expert feature engineering pipeline for datasets where the categorical data treatment was crucial for high performance. Operations that add new features based on existing categorical features (e.g., frequency encoding) are always considered part of the expert preprocessing pipeline, even though models like CatBoost use this information natively. Similarly, we remove

the treatment of missing values from the expert pipelines and leave that to the respective models whenever possible.

The following feature engineering techniques were applied most frequently over all datasets: groupby interactions (4), two-order categorical interactions (3), feature selection (3), categorical frequency encoding (3), dimensionality reduction (2), three-order categorical interaction (2), 2-order arithmetic interactions (2), sum of missing values in a row (2), and sum of zeros in a row (2). Details on all implemented techniques for the datasets can be found in our code.

**Feature Engineering Techniques used for test-time adaptation.** Of the abovementioned feature engineering techniques, the following were utilized as test-time feature engineering techniques:

- Counts of categorical features (AEAC, IFD, PSSDP)

- Dimensionality reduction (OGPCC, SCS)

- Groupby interactions (SCS, IFD)

- Occurrence of numeric features from train data in the test data (SCTP)

- Model-based Denoising/Smoothing by training an XGBoost model to predict features and using out-of-fold predictions as features (PSSDP)

**Cross-validation procedures.** We used the same CV split type for each dataset as the expert solutions but unified the number of folds across all our datasets. We used 10 folds for most datasets, as this worked well for all datasets. For classification tasks, the folds were stratified across the target. For the IFD dataset, we split the data based on the month the data was collected, which resulted in six folds. For faster training, fewer folds could be used for large datasets with similar results. For large datasets, most expert solutions used fewer folds, e.g., for the PSSDP competition.

### A.3    Discussion on Test-Time Feature Engineering

To deal with distribution shifts, test-time adaptation is a conceptual framework where the model parameters are allowed to depend on the test sample $x$ but not on its unknown label $y$. This matches the common ML competition setup, where test samples are given, but the target is hidden. We found that successful participants in Kaggle competitions often use the test data for feature engineering. Hence, we established that using test data for feature engineering in Kaggle competitions can be considered a special kind of test-time adaptation. This subsection discusses when this practice can be considered for real-world applications and when it is an unfair and unrealistic setup for a task.

We argue that the common ML setup allowing for test-time adaptation corresponds to a frequent real-world application scenario where 1) The data to predict arrives in batches, 2) No real-time predictions are required, 3) The train data is still available at test time, and 4) Retraining the model at test time is feasible. The first criterion is important as the employed test-time feature engineering techniques (e.g. dimensionality reduction and frequency encoding) often required the presence of many test samples at once. It is unclear whether this kind of domain adaptation would work per sample. The other criteria are necessary as test-time feature engineering always requires retraining the model. Consequently, test-time feature engineering is not applicable in online learning, if the number of test samples is small, or if the models are not retrainable (i.e., large-scale models). Importantly, while test-time feature engineering might be infeasible in such applications, other test-time adaptation techniques might still apply. One example of a task conceptualization amenable to test-time feature engineering is product return prediction, where samples are collected over a day, and a (lightweight) model can be retrained daily. In this scenario, using the test data in an unsupervised fashion for better adaptation to possible distribution shifts is feasible. After examining the application scenarios of the tasks in our framework, we found that most of them, like customer transaction prediction (SCTP) and customer satisfaction prediction (SCS), would allow such a setup, although likely with smaller amounts of test data than used for the competitions. Furthermore, our discussion in Subsection 4.4 reveals that many tabular datasets not in our scope have temporal components and thus may be amenable to test-time feature engineering.

# B    Experimental Details

In this Section, we discuss all aspects of our experiments that are not a part of our proposed evaluation framework but rather are design choices we made for our experiments.

## B.1    Software and Hardware

The deep learning models, CatBoost, and XGBoost, were trained using one or more of the following GPU hardware, depending on the availability: NVIDIA H100, NVIDIA A100, NVIDIA RTX A6000, or NVIDIA A40. LightGBM and AutoGluon were trained using the following CPU hardware: Intel(R) Xeon(R) CPU E2640v2 @ 2,00 GHz; Intel(R) Xeon(R) CPU E5-2640 v3 @ 2.60GHz.

## B.2    Model-Specific Preprocessing

For the tree-based models, model-specific preprocessing only included the correct assignment of datatypes to categorical features. For the deep learning models, the preprocessing was defined in line with the related work [25, 26]. For regression, the target is normalized to zero mean and unit variance. For numeric features, missing values are replaced with the mean, and the features are normalized using ScikitLearn's QuantileTransformer [58]. Categorical features are ordinally encoded as ResNet, FTTransformer, and MLP-PLR use embeddings for categorical features. The GRANDE library includes its own preprocessing, which contains the same steps as for the other deep learning models but uses leave-one-out-encoding for categorical features. For AutoGluon, all the preprocessing is left to the AutoML framework.

## B.3    Model Training and Hyperparameter Optimization

We use the optuna library [1] for hyperparameter optimization. Each model is optimized for 100 trials with the first 20 trials being random search trials and the remaining 80 using the multivariate Tree-structured Parzen Estimator algorithm [3, 17]. The models are trained using cross entropy for classification and mean squared error for regression. The AdamW optimizer is used for training the deep learning models [49]. Whenever possible, we use the task metric for validation during model training and for choosing the best hyperparameters. Instead of the R2 metric, we use rmse, as the objective is the same. Moreover, we we use rmse whenever the metric is rmsle, as we already transformed the target prior to fitting. Instead of Gini, we use AUC, as the metrics are convertible. For AutoGluon, we use the 'best_quality' preset configuration and a time limit of 10 hours. Everything else is left to the AutoML library itself. We try to use default hyperparameters and tuning ranges that have been shown to perform well for each model. For that, we orient on different related work [25, 26, 23, 55, 53] and the library documentations. Some of the datasets we use are quite large compared to most related work. Our goal was to evaluate each of the included models with an equal number of hyperparameter trials. Therefore, we did not use time budgets to constrain the number of trials per model and dataset. As this leads to long computation times for some models, we did not tune the representation capacity parameters for FTTransformer, as it was the most time-expensive model in our scope. All default hyperparameters and search spaces can be seen in Tables 6-12.

| Hyperparameter | Default | Search distribution |
|---|---|---|
| n_estimators | 4000 | - |
| patience | 200 | - |
| learning_rate | 0.3 | LogUniform[1e-3, 0.7] |
| max_depth | 6 | UniformInt[1, 11] |
| colsample_bytree | 1. | Uniform[0.5,1.] |
| subsample | 1. | Uniform[0.5,1.] |
| min_child_weight | 1. | LogUniform[1, 100] |
| reg_alpha | 0. | LogUniform[1e-8, 100] |
| reg_lambda | 1. | LogUniform[1, 4] |
| gamma | 0. | LogUniform[1e-8, 7] |

Table 6: Hyperparameter configurations for XGBoost.

| Hyperparameter | Default | Search distribution |
|---|---|---|
| iterations | 4000 | - |
| patience | 200 | - |
| learning_rate | 0.1 | LogUniform[1e-3, 0.7] |
| max_depth | -1 | {-1, UniformInt[1, 11]} |
| min_data_in_leaf | 20 | {20, 50, 100, 500, 1000, 2000} |
| num_leaves | 31 | UniformInt[2, 2047] |
| lambda_l2 | 0. | LogUniform[1e-4, 10.] |
| feature_fraction | 1. | Uniform[0.5, 1.] |
| bagging_fraction | 1. | Uniform[0.5, 1.] |
| min_sum_hessian_in_leaf | 1e-3 | LogUniform[1e-4,100.0] |

Table 7: Hyperparameter configurations for LightGBM. If max_depth>=1, the possible num_leaves ranges were adjusted to be in a space feasible with the respective depth.

| Hyperparameter | Default | Search Distribution |
|---|---|---|
| iterations | 4000 | - |
| od_type | "Iter" | - |
| od_wait | 200 | - |
| learning_rate | auto | LogUniform[1e-3, 1.] |
| max_depth | 6 | UniformInt[1, 11] |
| l2_leaf_reg | 3.0 | LogUniform[1,30] |
| bagging_temperature | 1 | Uniform[0,1] |

Table 8: Hyperparameter configurations for CatBoost. In the default setting, the library automatically determines a dataset-specific learning rate.

| Hyperparameter | Default | Search distribution |
|---|---|---|
| epochs | 200 | - |
| patience | 5 | - |
| batch_size | 128 | - |
| learning_rate | 1e-4 | LogUniform[1e-5, 1e-2] |
| weight_decay | 1e-5 | LogUniform[1e-6, 1e-3] |
| # Layers | 2 | UniformInt[1, 8] |
| Layer size | 192 | UniformInt[64, 1024] |
| Hidden factor | 2. | Uniform[1, 4] |
| Hidden dropout | 0.25 | Uniform[0., 0.5] |
| Residual dropout | 0. | Uniform[0., 0.5] |
| Categorical embedding size | 8 | UniformInt[4, 512] |

Table 9: Hyperparameter configurations for ResNet.

| Hyperparameter | Default | Search distribution |
|---|---|---|
| epochs | 200 | - |
| patience | 5 | - |
| batch_size | 128 | - |
| learning_rate | 1e-4 | LogUniform[1e-5, 1e-3] |
| weight_decay | 1e-5 | LogUniform[1e-6, 1e-3] |
| # Layers | 3 | - |
| Layer size | 192 | - |
| # Attention heads | 8 | - |
| Hidden factor | $\frac{4}{3}$ | - |
| Hidden dropout | 0.1 | Uniform[0., 0.5] |
| Attention dropout | 0.2 | Uniform[0., 0.5] |
| Residual dropout | 0. | Uniform[0., 0.2] |
| Categorical embedding size | 8 | - |

Table 10: Hyperparameter configurations for FTTransformer. Note that weight decay is only applied to some layers of the model. For details, see [25].

| Hyperparameter | Default | Search distribution |
|---|---|---|
| epochs | 200 | - |
| patience | 5 | - |
| batch_size | 128 | - |
| learning_rate | 1e-3 | LogUniform[5e-5, 5e-3] |
| weight_decay | 1e-4 | LogUniform[1e-6, 1e-3] |
| # Layers | 2 | UniformInt[1, 8] |
| Layer size | 192 | UniformInt[1, 1024] |
| Categorical embedding size | 8 | UniformInt[1, 128] |
| Numerical embedding size | 8 | UniformInt[1, 128] |
| Dropout | 0.25 | Uniform[0., 0.5] |
| frequency_init_scale | 0. | LogUniform[1e-2, 10.] |

Table 11: Hyperparameter configurations for MLP-PLR.

| Hyperparameter | Default | Search distribution |
|---|---|---|
| epochs | 1000 | - |
| patience | 25 | - |
| batch_size | 64 | - |
| depth | 5 | - |
| n_estimators | 2048 | - |
| learning_rate_weights | 0.005 | LogUniform[1e-4, 0.25] |
| learning_rate_index | 0.01 | LogUniform[1e-4, 0.25] |
| learning_rate_values | 0.01 | LogUniform[1e-4, 0.25] |
| learning_rate_leaf | 0.01 | LogUniform[1e-4, 0.25] |
| cosine_decay_steps | 0 | {0, 100, 1000} |
| dropout | 0.0 | {0, 0.25} |
| selected_variables | 0.8 | {0.5, 0.75, 1.} |
| Focal loss | 0.0 | {False, True} |
| Temperature | 0.0 | {0, 0.25} |

Table 12: Hyperparameter configurations for GRANDE. The focal loss and temperature parameters only apply to classification tasks.

# C    Detailed Performance Results

In this Section, we provide the leaderboard position results for all the experiments in the main paper, separated by hyperparameter regime and preprocessing pipeline. The results can be seen in Tables 13, 14, 15, and 16.

|  |  | XGBoost | LightGBM | CatBoost | ResNet | FTT | MLP-PLR | GRANDE |
|---|---|---|---|---|---|---|---|---|
| Default | MBGM | 0.17 | 0.226 | **0.997** | 0.231 | 0.267 | 0.605 | 0.143 |
|  | SVPC | 0.799 | **0.929** | 0.889 | 0.798 | 0.798 | **0.92** | 0.795 |
|  | AEAC | 0.553 | 0.613 | **0.91** | 0.503 | 0.544 | 0.527 | 0.43 |
|  | OGPCC | **0.819** | 0.803 | 0.795 | 0.706 | 0.729 | 0.776 | 0.69 |
|  | SCS | 0.466 | 0.439 | 0.469 | 0.368 | **0.6** | 0.412 | 0.37 |
|  | BPCCM | 0.256 | 0.28 | **0.953** | 0.261 | 0.281 | 0.31 | 0.07 |
|  | SCTP | 0.338 | 0.364 | **0.431** | 0.287 | 0.374 | 0.315 | 0.376 |
|  | HQC | 0.319 | 0.343 | **0.936** | 0.378 | 0.47 | 0.418 | 0.455 |
|  | IFD | 0.311 | 0.324 | **0.519** | 0.226 | 0.408 | 0.294 | 0.177 |
|  | PSSDP | 0.288 | 0.301 | **0.519** | 0.315 | 0.478 | 0.258 | 0.347 |
| Light HPO | MBGM | 0.552 | 0.312 | **0.998** | 0.272 | 0.39 | 0.708 | 0.649 |
|  | SVPC | 0.929 | **0.937** | 0.895 | 0.798 | 0.798 | **0.946** | 0.925 |
|  | AEAC | 0.544 | 0.693 | **0.945** | 0.693 | 0.614 | 0.56 | 0.474 |
|  | OGPCC | 0.834 | **0.888** | 0.799 | 0.712 | 0.748 | 0.808 | 0.587 |
|  | SCS | 0.609 | 0.543 | 0.557 | 0.374 | **0.993** | 0.529 | 0.4 |
|  | BPCCM | 0.578 | 0.398 | **0.978** | 0.285 | 0.31 | 0.357 | 0.185 |
|  | SCTP | 0.448 | 0.392 | 0.448 | 0.297 | 0.401 | **0.501** | 0.4 |
|  | HQC | 0.865 | 0.414 | **0.987** | 0.378 | 0.491 | 0.527 | 0.509 |
|  | IFD | 0.461 | 0.525 | **0.54** | 0.22 | 0.334 | 0.267 | 0.201 |
|  | PSSDP | 0.583 | 0.392 | 0.555 | 0.313 | 0.493 | **0.656** | 0.407 |
| Extensive HPO | MBGM | 0.476 | 0.503 | **0.999** | 0.334 | 0.448 | 0.8 | 0.615 |
|  | SVPC | 0.932 | **0.946** | 0.917 | 0.798 | 0.798 | **0.947** | 0.932 |
|  | AEAC | 0.585 | 0.687 | **0.953** | 0.691 | 0.669 | 0.6 | 0.474 |
|  | OGPCC | **0.887** | **0.896** | 0.845 | 0.724 | 0.742 | 0.878 | 0.776 |
|  | SCS | 0.692 | 0.73 | 0.542 | 0.351 | **0.945** | 0.478 | 0.427 |
|  | BPCCM | 0.587 | 0.499 | **0.986** | 0.301 | 0.333 | 0.362 | 0.185 |
|  | SCTP | **0.51** | 0.428 | 0.495 | 0.298 | 0.408 | **0.518** | 0.496 |
|  | HQC | 0.911 | 0.409 | **0.991** | 0.414 | 0.527 | 0.527 | 0.619 |
|  | IFD | 0.533 | **0.662** | 0.552 | 0.223 | 0.518 | 0.268 | 0.245 |
|  | PSSDP | **0.656** | 0.463 | 0.586 | 0.308 | 0.549 | **0.656** | 0.418 |

Table 13: Leaderboard position of models trained with varying hyperparameter optimization regimes on datasets after standardized preprocessing. The best model (+/- 0.01) is highlighted.

|  |  | XGBoost | LightGBM | CatBoost | ResNet | FTT | MLP-PLR | GRANDE |
|---|---|---|---|---|---|---|---|---|
| Default | MBGM | 0.706 | 0.545 | **0.999** | 0.626 | 0.641 | 0.964 | 0.61 |
|  | SVPC | **0.993** | **0.987** | **0.987** | 0.941 | 0.968 | **0.989** | 0.946 |
|  | AEAC | 0.736 | 0.84 | **0.937** | 0.695 | 0.914 | 0.407 | 0.407 |
|  | OGPCC | **0.806** | 0.792 | **0.797** | 0.702 | 0.718 | 0.731 | 0.681 |
|  | SCS | 0.59 | 0.758 | 0.673 | 0.366 | **0.879** | 0.466 | 0.393 |
|  | BPCCM | **0.994** | **0.995** | **0.994** | 0.987 | **0.993** | **0.995** | **0.991** |
|  | SCTP | 0.311 | 0.373 | **0.417** | 0.284 | 0.376 | 0.345 | 0.395 |
|  | HQC | 0.354 | 0.371 | **0.953** | 0.46 | **0.948** | 0.393 | 0.368 |
|  | IFD | **0.83** | 0.775 | 0.775 | 0.215 | 0.741 | 0.615 | 0.462 |
|  | PSSDP | 0.511 | 0.479 | 0.743 | 0.531 | **0.993** | 0.361 | 0.377 |
| Light HPO | MBGM | **0.991** | 0.794 | **0.999** | 0.727 | 0.774 | 0.908 | 0.985 |
|  | SVPC | **0.993** | **0.992** | 0.987 | 0.951 | 0.971 | **0.987** | **0.988** |
|  | AEAC | 0.734 | 0.932 | **0.945** | 0.905 | 0.78 | 0.693 | 0.495 |
|  | OGPCC | 0.823 | **0.852** | 0.803 | 0.705 | 0.748 | 0.839 | 0.563 |
|  | SCS | 0.824 | 0.754 | 0.783 | 0.382 | **0.837** | 0.555 | 0.519 |
|  | BPCCM | **0.993** | **0.991** | **0.995** | **0.99** | **0.993** | **0.995** | **0.995** |
|  | SCTP | 0.413 | 0.418 | **0.483** | 0.288 | 0.37 | **0.483** | 0.438 |
|  | HQC | **0.989** | 0.501 | **0.989** | 0.45 | **0.986** | 0.973 | 0.57 |
|  | IFD | **0.988** | 0.963 | 0.836 | 0.2 | 0.71 | 0.572 | 0.744 |
|  | PSSDP | 0.716 | 0.708 | 0.735 | 0.701 | **0.94** | 0.741 | 0.575 |
| Extensive HPO | MBGM | 0.95 | 0.859 | **0.994** | 0.71 | 0.875 | 0.934 | 0.945 |
|  | SVPC | **0.993** | **0.992** | 0.987 | 0.949 | 0.978 | **0.987** | **0.985** |
|  | AEAC | 0.762 | **0.937** | **0.928** | 0.832 | 0.777 | 0.702 | 0.525 |
|  | OGPCC | **0.867** | 0.856 | 0.842 | 0.714 | 0.751 | **0.871** | 0.777 |
|  | SCS | **0.953** | 0.941 | 0.777 | 0.377 | 0.702 | 0.627 | 0.711 |
|  | BPCCM | **0.992** | **0.992** | **0.996** | **0.991** | **0.992** | **0.992** | **0.996** |
|  | SCTP | 0.521 | 0.5 | 0.557 | 0.293 | 0.376 | 0.499 | **0.962** |
|  | HQC | **0.99** | 0.487 | **0.991** | 0.468 | **0.986** | **0.982** | 0.839 |
|  | IFD | **0.988** | **0.985** | 0.809 | 0.216 | 0.665 | 0.62 | 0.736 |
|  | PSSDP | **0.994** | 0.944 | 0.973 | 0.684 | **0.99** | **0.99** | 0.605 |

Table 14: Leaderboard position of models trained with varying hyperparameter optimization regimes on datasets after feature engineering. The best model (+/- 0.01) is highlighted.

|  |  | XGBoost | LightGBM | CatBoost | ResNet | FTT | MLP-PLR | GRANDE |
|---|---|---|---|---|---|---|---|---|
| Default | AEAC | 0.944 | 0.948 | **0.98** | 0.725 | 0.758 | 0.46 | 0.445 |
|  | OGPCC | **0.856** | **0.847** | 0.84 | 0.714 | 0.714 | 0.783 | 0.718 |
|  | SCS | 0.543 | 0.611 | 0.701 | 0.39 | **0.892** | 0.635 | 0.431 |
|  | SCTP | **0.985** | **0.987** | **0.988** | 0.302 | **0.983** | **0.986** | **0.988** |
|  | IFD | **0.986** | **0.983** | 0.913 | 0.214 | 0.774 | 0.53 | 0.533 |
|  | PSSDP | 0.508 | 0.491 | 0.746 | 0.59 | **0.981** | 0.315 | 0.374 |
| Light HPO | AEAC | 0.95 | 0.96 | **0.99** | 0.932 | 0.943 | 0.776 | 0.507 |
|  | OGPCC | 0.894 | **0.915** | 0.852 | 0.731 | 0.75 | 0.842 | 0.566 |
|  | SCS | **0.937** | 0.778 | 0.773 | 0.425 | 0.904 | 0.611 | 0.485 |
|  | SCTP | **0.988** | **0.988** | **0.989** | 0.315 | **0.985** | **0.991** | **0.989** |
|  | IFD | **0.991** | **0.989** | **0.987** | 0.211 | 0.692 | 0.642 | 0.646 |
|  | PSSDP | 0.94 | 0.773 | 0.792 | 0.745 | **0.978** | 0.707 | 0.609 |
| Extensive HPO | AEAC | 0.953 | 0.961 | **0.991** | 0.922 | 0.932 | 0.932 | 0.534 |
|  | OGPCC | **0.922** | **0.923** | 0.884 | 0.72 | 0.78 | 0.878 | 0.808 |
|  | SCS | **0.975** | 0.842 | 0.904 | 0.362 | 0.798 | 0.734 | 0.798 |
|  | SCTP | **0.991** | **0.99** | **0.991** | 0.346 | **0.985** | **0.992** | **0.991** |
|  | IFD | **0.992** | **0.992** | 0.972 | 0.204 | 0.739 | 0.647 | 0.708 |
|  | PSSDP | **0.992** | 0.982 | **0.99** | 0.741 | **0.994** | **0.995** | 0.651 |

Table 15: Leaderboard position of models trained with varying hyperparameter optimization regimes on datasets after test-time feature engineering as a preprocessing method for test-time adaptation. The best model (+/- 0.01) is highlighted.

|      | MBGM | SVPC | AEAC | OGPCC | SCS | BPCCM | SCTP | HQC | IFD | PSSDP |
|------|------|------|------|-------|-----|-------|------|-----|-----|-------|
| Def. | 0.74 | 0.799 | 0.618 | **0.996** | **0.92** | **0.991** | 0.498 | **0.992** | 0.205 | 0.562 |
| FE   | **0.964** | **0.963** | 0.953 | 0.983 | **0.999** | **0.995** | 0.531 | **0.992** | 0.351 | 0.707 |
| TTA  | -    | -    | 0.993 | 0.995 | **1.0** | -     | 0.991 | -   | **0.432** | **0.742** |

Table 16: Leaderboard position of AutoGluon on the private Kaggle leaderboard after different preprocessing applied. The best results (+/- 0.01) are highlighted.

# D Additional Results

## D.1 Runtime analysis

As our experiments were conducted with varying hardware setups depending on availability, we cannot directly compare runtimes. Hence, we conduct a separate experiment where we train each model with default hyperparameters on each dataset and preprocessing pipeline on the same hardware. In particular, we train all models in a setup where all folds are trained in parallel distributed equally over two NVIDIA RTX A6000 GPUs. This setup is particularly beneficial for small datasets as well as for MLP-like neural architectures as sequential training does not fully utilize the capabilities of modern GPUs on these datasets and models. Figure 7 shows that XGBoost is the only Pareto point after feature engineering and with test-time adaptation as it is the fastest and best performing model at the same time. For the standardized setup the tree-based models build the Pareto frontier. However, as in this plot runtime is measured with default hyperparameters and performance with extensively tuned hyperparameters, the results do not represent the actual time required to obtain the performance results.

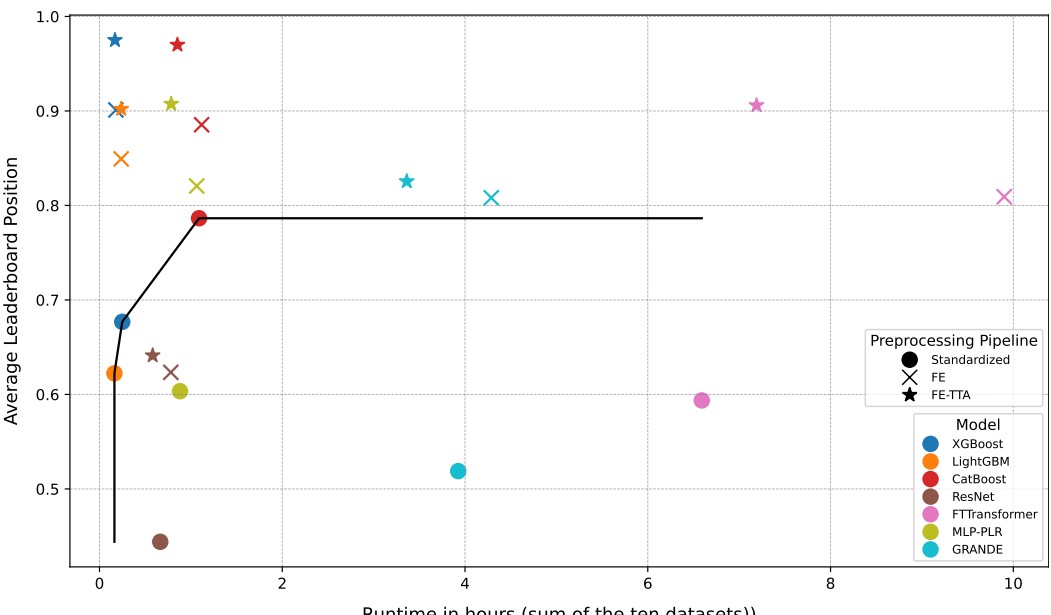

Figure 7: Runtime analysis. Performance is reported as the leaderboard position after extensive hyperparameter optimization averaged over the ten datasets. Time is reported as the total time in minutes required to train a model with default hyperparameters on all ten datasets with a particular preprocessing pipeline. Note that each experiment was conducted with parallelizing the training of folds, s.t. the time in GPU hours is much higher than the reported time. The black line represents the Pareto frontier for the models trained in the standardized preprocessing pipeline. For the other pipelines, XGBoost was the only Pareto optimum.

## D.2  Model Ranking Variances Across Dataset

Figure 17 shows that the dataset-variance is rather high, mainly because the hardness of the task differs among datasets, especially with standardized preprocessing. It can be seen that the variance of the best performing methods strongly decreases after TTA, as good preprocessing eases the prediction tasks for the previously hard datasets.

|  | CatBoost | XGBoost | LightGBM | MLP-PLR | FTTransformer | GRANDE | ResNet |
|---|---|---|---|---|---|---|---|
| Stand. | 0.79 (0.21) | 0.68 (0.17) | 0.62 (0.19) | 0.6 (0.22) | 0.59 (0.19) | 0.52 (0.23) | 0.44 (0.21) |
| FE | 0.89 (0.14) | 0.9 (0.15) | 0.85 (0.19) | 0.82 (0.19) | 0.81 (0.2) | 0.81 (0.17) | 0.62 (0.27) |
| TTA | 0.97 (0.04) | 0.97 (0.03) | 0.9 (0.16) | 0.91 (0.12) | 0.91 (0.1) | 0.83 (0.16) | 0.64 (0.28) |

Table 17: Average leaderboard position with variance per dataset and pipeline.

## D.3  Comparison of Preprocessing Pipelines per Model and Dataset

Figure 8 visualizes the comparison of different pipelines per model and dataset. It can be seen that almost all models benefit from feature engineering and test-time adaptation on almost all datasets. The only remarkable outlier is FTTransformer on the SCS dataset. The otherwise consistent results support our claim that feature engineering and test-time adaptation are important components of tabular machine learning competitions.

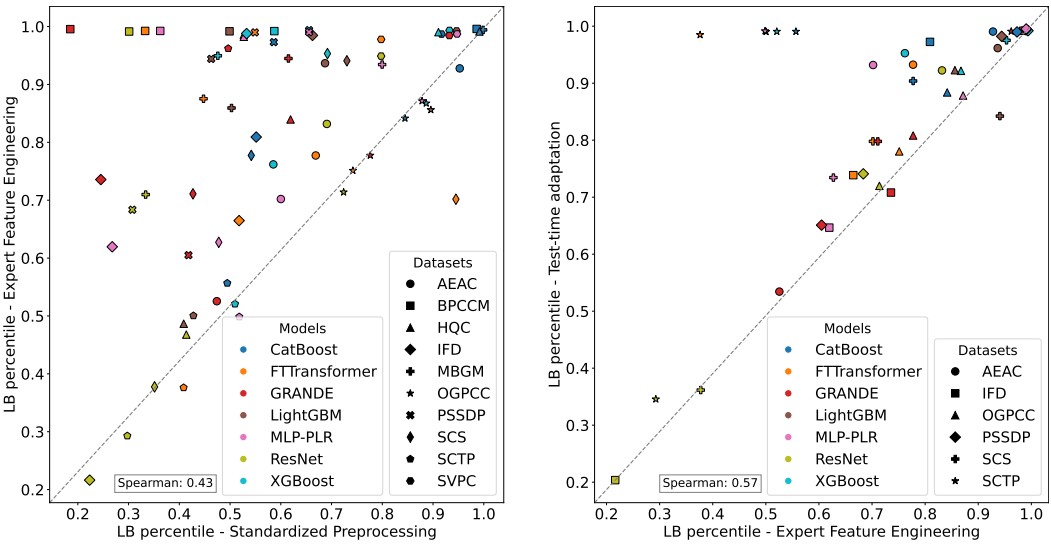

Figure 8: Leaderboard positions of models and datasets in different preprocessing pipelines.

## D.4  Analysis of Modeling Components

In this Subsection we complement our analysis of modeling components in the main paper with additional analyses from different perspectives. Figure 9 shows the distributions of the leaderboard positions of all our experiments, grouped by different modeling components. It can be seen that without expert feature engineering, most submissions are far from the top percentiles on the leaderboard, while after expert feature engineering, most submissions score top ranks. After test-time adaptation, the density of top submissions increases even more. Moreover, the importance of hyperparameter optimization can be seen. Regarding model selection, CatBoost clearly dominates, mainly due to the property of achieving robustly strong results with default hyperparameters.

These results are in line with common practice in ML competitions [68]. While recent work strongly focuses on model selection [26, 55], participants of Kaggle competitions typically stick to few model classes and instead focus on developing feature engineering techniques for these particular models [35]. Predictive Machine Learning is a winner-takes-all game. Selecting another model than the one

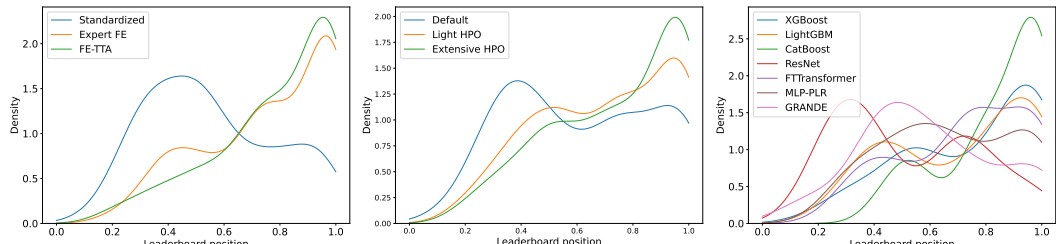

Figure 9: Kernel Density Estimation of all results grouped by different modeling components (Left: Preprocessing Pipelines, Center: HPO regimes, Right: Models).

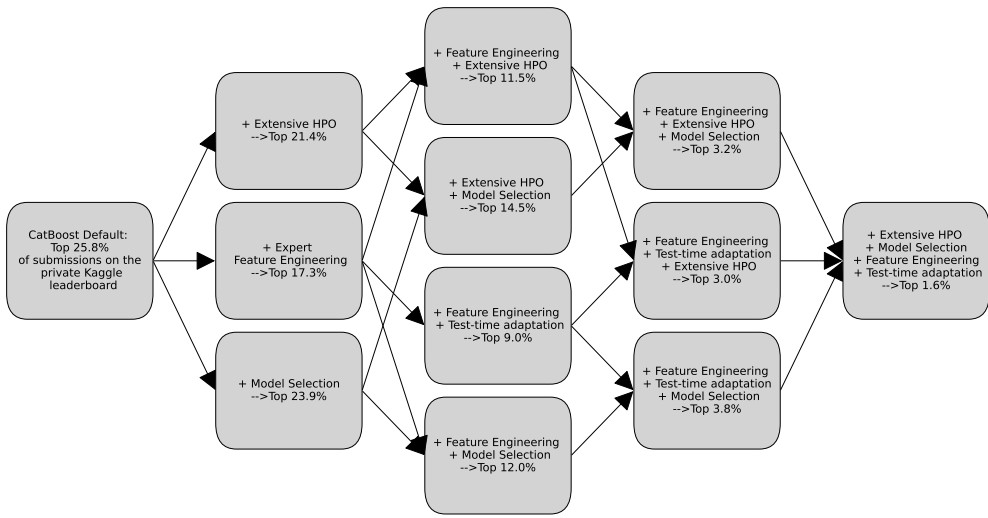

Figure 10: Average Gains from different modeling choices from a winner-takes-all perspective with CatBoost as the default model. Lower values mean a higher leaderboard position (unlike the rest of the paper).

that is known to work best is only important if the default model fails or if ensembling is necessary. Hence, it makes sense to look at the problem from the winner-takes-all perspective and to evaluate performance gains from different modeling decisions w.r.t. a strong baseline. It is known from related work that CatBoost is the strongest model with default hyperparameters [55]. Hence, we evaluate performance gains over a CatBoost baseline. Figure 10 illustrates average leaderboard position gains over the baseline for different modeling decisions. It can be seen that without feature engineering, the best average leaderboard position is the 14.5% percentile, while without model selection, it is the 3% percentile. Hence, one of the most important takeaways is that current tabular ML research overemphasizes model evaluation but underestimates data preprocessing especially feature engineering. While TTA increases the average leaderboard position by 8.3% in the default setting, its importance after model selection and hyperparameter optimization reduces. Without TTA, the best achievable average leaderboard position was 3.2% and 1.6% with TTA, indicating a relatively small but important effect. In this Figure, the average top position is the 1.6% percentile because we only considered single models. The missing component for scoring in the top 1% percentile is ensembling, which we achieve using AutoGluon in the main paper.

To statistically test our results, we estimate the effect of different modeling components in a mixed-effects regression analysis. We use the leaderboard position of all our experiments as the target variable. The samples are all our experiments with leaderboard evaluations resulting from all dataset-preprocessing-model-HPO combinations. To control for different dataset difficulty, we use the dataset as a random effect. The fixed effects are:

- *Featue Engineering* {0, 1}: 1 if feature engineering (with and without TTA) was applied;
- *Test-Time Adaptation* {0, 1}: 1 if test-time adaptation was applied;

- *Model Selection* {-1, 0, 1}: -1 if CatBoost is the model, 1 if the model is the best of all models on a dataset-preprocessing-HPO combination;
- *Light HPO* {0,1}: 1 if light HPO was applied;
- *Extensive HPO* {0,1}: 1 if extensive HPO was applied;

The results in Table 18 confirm the strong overall importance of feature engineering and the relevance of HPO and test-time adaptation. Furthermore, it can be seen that using a model other than CatBoost does not lead to significant gains on average. We want to emphasize that while this reflects the general trend across all experiments, for some constellations, other models achieve strong gains over CatBoost.

| Model: | MixedLM | Dependent Variable: | leaderboard position |
|---|---|---|---|
| No. Observations: | 546 | Method: | REML |
| No. Groups: | 10 | Scale: | 0.0380 |
| Min. group size: | 42 | Log-Likelihood: | 88.5557 |
| Max. group size: | 63 | Converged: | Yes |
| Mean group size: | 54.6 | | |

| | Coef. | Std.Err. | z | P> \|z\| | [0.025 | 0.975] |
|---|---|---|---|---|---|---|
| Intercept | 0.485 | 0.041 | 11.898 | 0.000 | 0.405 | 0.565 |
| Feature Engineering | 0.201 | 0.019 | 10.561 | 0.000 | 0.164 | 0.238 |
| Test-Time Adaptation | 0.080 | 0.023 | 3.437 | 0.001 | 0.034 | 0.126 |
| Model Selection | 0.004 | 0.016 | 0.240 | 0.810 | -0.027 | 0.034 |
| Light HPO | 0.085 | 0.020 | 4.163 | 0.000 | 0.045 | 0.125 |
| Extensive HPO | 0.125 | 0.020 | 6.137 | 0.000 | 0.085 | 0.165 |
| Dataset (Group Variable) | 0.013 | 0.035 | | | | |

Table 18: Mixed Linear Model Regression Results

### D.5 Evaluation At Different Snapshots of the Competitions

In the main paper, we evaluated results w.r.t. the end of the competition as a reference snapshot. All publicly available metadata from Kaggle competitions is available at Meta Kaggle,(`https://www.kaggle.com/datasets/kaggle/meta-kaggle`). This allows us to additionally use different points in time when the competition took place. Table 19 shows the performance of the best model in the standardized preprocessing pipeline after varying number of days in the competition. For most datasets, scoring high positions at the first day without feature engineering is possible with our framework. This shows that our framework can be of use for participants in Kaggle competitions to obtain first baseline results.

### D.6 Using Our Framework for Evaluating New Methods

In Section 5 we identified directions for future work. These directions were based on general insights of our analysis for the tabular data field. This subsection will showcase how our framework can be utilized to develop new approaches and compare them to expert solutions. Our framework contains, to our knowledge, the largest collection of implemented expert solutions for relevant datasets. Hence, our framework is especially useful to researchers developing AutoML solutions, especially focusing on feature engineering. Furthermore, our framework can be useful to researchers developing model-specific and data-agnostic preprocessing pipelines, i.e., for novel neural networks. In addition, our framework can be used to develop test-time adaptation methods for tabular data. Figure 11 shows four particular challenges for future work in tabular Deep Learning and AutoML for which our framework can serve as a benchmark to measure progress: A) Develop a neural network not relying on feature engineering techniques; B) Develop a neural network capable of test-time adaptation to replace the often infeasible test-time feature engineering. C) Create a universal model-agnostic automated feature engineering pipeline that surpasses expert feature engineering; D) Enhance AutoML solutions to outperform expert modeling pipelines;

|  |  | 1 | 7 | 14 | 21 | 28 | 45 | 60 | end |
|---|---|---|---|---|---|---|---|---|---|
| MBGM | Best single | 1.0 | 0.998 | 0.997 | 0.996 | 0.997 | 0.994 | 0.994 | 0.994 |
|  | AutoGluon | 0.87 | 0.814 | 0.706 | 0.67 | 0.653 | 0.616 | 0.616 | 0.616 |
| SVPC | Best single | 1.0 | 1.0 | 1.0 | 1.0 | 1.0 | 0.991 | 0.964 | 0.946 |
|  | AutoGluon | 1.0 | 1.0 | 1.0 | 1.0 | 1.0 | 0.985 | 0.873 | 0.794 |
| AEAC | Best single | 1.0 | 1.0 | 0.996 | 0.991 | 0.989 | 0.982 | 0.958 | 0.953 |
|  | AutoGluon | 0.98 | 0.894 | 0.781 | 0.739 | 0.698 | 0.667 | 0.638 | 0.616 |
| OGPCC | Best single | 0.996 | 0.994 | 0.987 | 0.984 | 0.977 | 0.963 | 0.908 | 0.896 |
|  | AutoGluon | 1.0 | 1.0 | 1.0 | 0.999 | 1.0 | 0.998 | 0.996 | 0.996 |
| SCS | Best single | 1.0 | 0.978 | 0.972 | 0.95 | 0.905 | 0.865 | 0.825 | 0.825 |
|  | AutoGluon | 1.0 | 0.972 | 0.966 | 0.936 | 0.885 | 0.837 | 0.792 | 0.792 |
| BPCCM | Best single | 1.0 | 0.998 | 0.996 | 0.997 | 0.997 | 0.993 | 0.991 | 0.986 |
|  | AutoGluon | 1.0 | 1.0 | 0.998 | 0.997 | 0.997 | 0.997 | 0.995 | 0.992 |
| SCTP | Best single | 0.991 | 0.895 | 0.893 | 0.865 | 0.681 | 0.551 | 0.511 | 0.511 |
|  | AutoGluon | 0.986 | 0.867 | 0.865 | 0.822 | 0.655 | 0.531 | 0.493 | 0.493 |
| HQC | Best single | 1.0 | 1.0 | 1.0 | 1.0 | 0.998 | 0.999 | 0.995 | 0.992 |
|  | AutoGluon | 1.0 | 1.0 | 1.0 | 1.0 | 1.0 | 1.0 | 0.995 | 0.993 |
| IFD | Best single | 1.0 | 0.991 | 0.985 | 0.978 | 0.922 | 0.776 | 0.694 | 0.621 |
|  | AutoGluon | 0.14 | 0.145 | 0.165 | 0.185 | 0.187 | 0.178 | 0.17 | 0.192 |
| PSSDP | Best single | 0.995 | 0.969 | 0.889 | 0.845 | 0.781 | 0.696 | 0.638 | 0.638 |
|  | AutoGluon | 0.968 | 0.889 | 0.786 | 0.727 | 0.667 | 0.593 | 0.552 | 0.552 |

Table 19: Private leaderboard position at different points in time of the competitions. For each competition, the performance of the best model trained in the standardized preprocessing pipeline is reported. The columns represent days after the competition started.

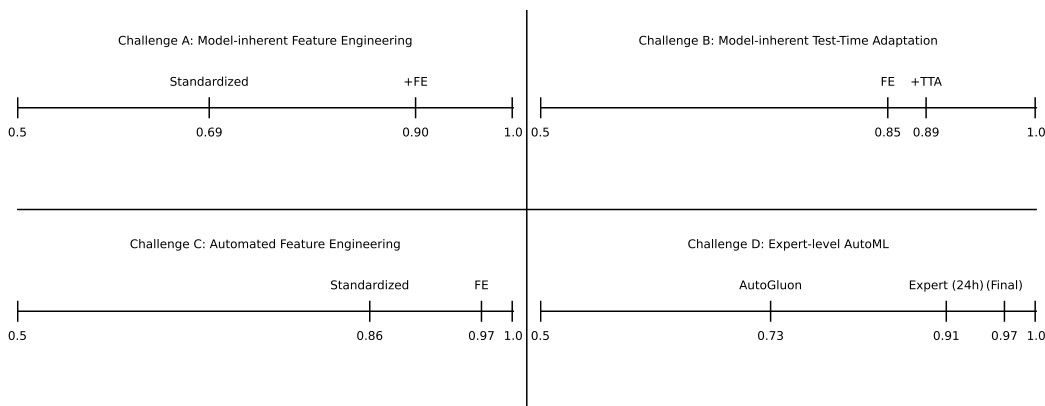

Figure 11: Challenges for further automating deep learning and AutoML for tabular data. Challenge A compares the best neural networks within the standardized and feature engineering pipelines after extensive HPO. Challenge B compares the best neural networks within the feature engineering and the test-time adaptation pipeline after extensive HPO. Challenge C compares the best model within the standardized pipeline to the best model within the feature engineering pipeline. Challenge D compares AutoGluon to the best submission after 24 hours and the best models within the feature engineering pipeline.

# E   Discussion of Limitations

The main goal of our experiments was to showcase the limitations of evaluation frameworks currently prevalent in tabular Machine Learning. This required extensive experiments in different preprocessing pipelines. Some limitations arising from this scope are:

- We use Kaggle competitions in an effort to evaluate more realistic tasks than in related work. It is important to highlight that the competition setup on Kaggle does not always reflect real-world tasks. However, due to the involvement of companies and institutions and the poor availability of high-quality tabular datasets [41], these are arguably among the most realistic datasets available as open-source data. Furthermore, one of our contributions was

to separate aspects from the main learning task that made competitions unrealistic (i.e., data leaks or, for some applications, test-time adaptation). This additionally improves the real-world transferability of our experiments.

- We split the overall expert preprocessing into feature engineering and test-time adaptation. However, pipelines could be differentiated further, or single-feature engineering techniques could be investigated. For instance, we could separate the expert feature engineering pipeline by whether expert feature selection is applied. However, due to the extent of our experiments, we focus on a pipeline perspective and leave fine-grained analyses of specific techniques for future work.

- It is worth mentioning that the implemented feature engineering steps for each dataset were always extracted from modeling pipelines were they worked well in combination with specific models, mostly tree-based. It might be the case that some models would additionally benefit from other preprocessing techniques that were not part of the implemented pipeline. A general observation was that feature engineering techniques working well for one model often also work well for others. Only for one dataset, the expert solution (slightly) differentiated between tree-based and neural network preprocessing.

- We use the leaderboard percentile as the main evaluation measure to have an external reference for the top performance on a dataset. A possible issue of that design choice is that the leaderboard of each dataset is differently distributed. Hence, what appears to be a large jump on a dataset might actually only be a small increase on the metric, while for another dataset, the same leaderboard increase might amount to a substantial increase in performance. However, averaging over datasets has a natural interpretation when using the leaderboard position, which is not there when using normalized versions of entirely different metrics. In addition to the evaluation in the main paper, we include evaluations on the original metrics in Appendix F. The results indicate that our claims similarly hold when evaluating using the original metrics.

- Another possible issue with using the leaderboard as a metric is the suggestive wording possibly leading to misinterpretations of what expert solutions in our context are. First, not all submitted solutions are expert solutions and the leaderboard is skewed. Users of our framework should avoid formulations such as "beating 99% of experts", as it is unclear which of the submitted solutions can be considered expert level. Instead, factually correct statements such as "top 1% of all competition participants" should be used.

- Our evaluation framework does not allow to assess whether one model is generally better than another model. We only claim that model comparisons change and that feature engineering and preprocessing greatly influence model comparison on our datasets. For a more generalizable model comparison using more datasets, we refer to related work [26].

- Due to the extent of our experiments (over 200,000 trained models), it was infeasible to repeat the experiments multiple times to obtain error bars. Nevertheless, our experiments include randomness (e.g., CV splits, weight initialization for deep learning models, or bagging for the tree-based models), limiting the generalizability of our results. However, the extent of our experiments and the clear differences between the implemented preprocessing pipelines over multiple models and datasets make the risk of randomness affecting our main claims very low despite not being explicitly quantified for all models.

- Due to the focus on incentivized Kaggle competitions, most datasets are from the finance domain and from North America or Europe. Hence, non-profit domains and other continents are underrepresented. To mitigate this, our analysis could be extended through competitions on other platforms such as Zindi [80]. However, as we wanted our framework to be easy to use, we focused on Kaggle, which contains an API for effortlessly downloading datasets and submitting predictions.

## F   Evaluation on the Original Task Metrics

This Section lists the main results using the original task metrics for all experiments. An overview of important components per dataset and model can be seen in Figure 12. All experimental results on the original metrics can be seen in Tables 20, 21, 22, and 23. Further results can be seen in our code. Overall, the evaluation using the original metrics aligns with our main findings.

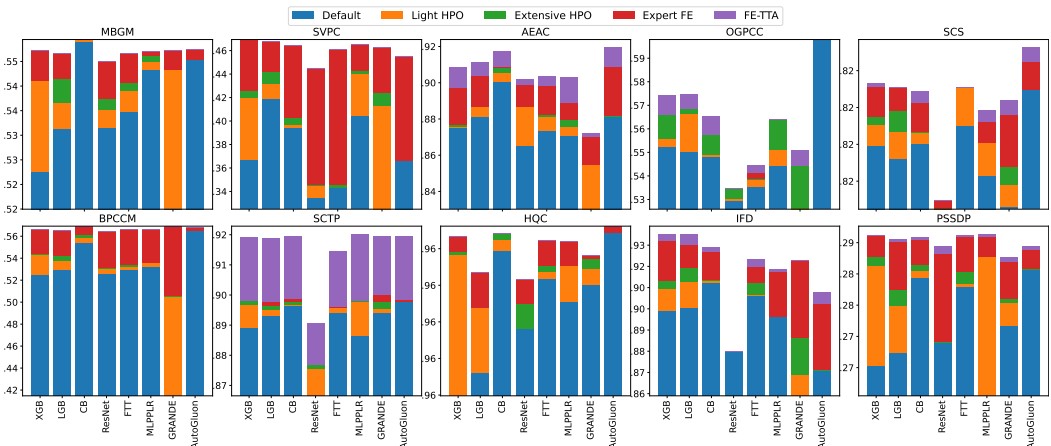

Figure 12: Performance gains from different modeling components on the original metrics of the Kaggle competitions. Higher values correspond to better performance. The original metric was reversed for SVPC, OGPCC, and BPCCM to align with the higher-is-better notation. 'Default' corresponds to the model performance with default hyperparameters in a standardized preprocessing pipeline. Light and extensive HPO correspond to tuning hyperparameters in the same preprocessing pipeline. Expert FE and FE-TTA correspond to the model performance with extensively tuned hyperparameters in the feature engineering and the test-time adaptation pipeline respectively.

|  |  | XGBoost | LightGBM | CatBoost | ResNet | FTT | MLP-PLR | GRANDE |
|---|---|---|---|---|---|---|---|---|
| Default | MBGM | 0.5276 | 0.5363 | **0.554** | 0.5366 | 0.5398 | 0.5483 | 0.52 |
|  | SVPC | 0.3666 | **0.419** | 0.3945 | 0.3348 | 0.3433 | 0.4041 | 0.3249 |
|  | AEAC | 0.8754 | 0.881 | **0.9005** | 0.8653 | 0.8738 | 0.8707 | 0.8302 |
|  | OGPCC | **0.5525** | 0.5501 | 0.5482 | 0.5293 | 0.5356 | 0.5443 | 0.5259 |
|  | SCS | 0.8239 | 0.8232 | 0.824 | 0.8205 | **0.825** | 0.8223 | 0.8206 |
|  | BPCCM | 0.525 | 0.5295 | **0.5539** | 0.5262 | 0.5295 | 0.5325 | 0.4148 |
|  | SCTP | 0.8892 | 0.893 | **0.8965** | 0.8666 | 0.894 | 0.8863 | 0.8942 |
|  | HQC | 0.96 | 0.9612 | **0.9679** | 0.9636 | 0.9664 | 0.9651 | 0.966 |
|  | IFD | 0.8989 | 0.9006 | **0.9121** | 0.8799 | 0.9065 | 0.8965 | 0.859 |
|  | PSSDP | 0.2703 | 0.2724 | **0.2843** | 0.274 | 0.283 | 0.2655 | 0.2766 |
| Light | MBGM | 0.5474 | 0.5416 | **0.5543** | 0.5402 | 0.5441 | 0.5499 | 0.5489 |
|  | SVPC | 0.4201 | 0.4316 | 0.3972 | 0.3454 | 0.343 | **0.4405** | 0.4129 |
|  | AEAC | 0.8738 | 0.8873 | **0.9056** | 0.8874 | 0.8811 | 0.8757 | 0.8548 |
|  | OGPCC | 0.5556 | **0.5664** | 0.5491 | 0.5305 | 0.5404 | 0.551 | 0.4905 |
|  | SCS | 0.8251 | 0.8247 | 0.8248 | 0.8208 | **0.8277** | 0.8245 | 0.8218 |
|  | BPCCM | 0.5436 | 0.538 | **0.5584** | 0.53 | 0.5325 | 0.5354 | 0.5057 |
|  | SCTP | 0.8968 | 0.8951 | 0.8968 | 0.8756 | 0.8956 | **0.8978** | 0.8956 |
|  | HQC | **0.9677** | 0.965 | **0.9685** | 0.9636 | 0.9667 | 0.967 | 0.9669 |
|  | IFD | 0.9097 | **0.9126** | **0.9135** | 0.8782 | 0.9012 | 0.8923 | 0.8688 |
|  | PSSDP | 0.2863 | 0.2799 | 0.2855 | 0.2739 | 0.2835 | **0.2878** | 0.2805 |
| Extensive | MBGM | 0.5461 | 0.5465 | **0.5545** | 0.5425 | 0.5457 | 0.5513 | 0.5484 |
|  | SVPC | 0.4256 | **0.4421** | 0.4024 | 0.3453 | 0.3455 | **0.4428** | 0.4239 |
|  | AEAC | 0.8771 | 0.8868 | **0.9086** | 0.887 | 0.8825 | 0.8793 | 0.8546 |
|  | OGPCC | 0.5662 | **0.5685** | 0.5575 | 0.5345 | 0.5387 | 0.5638 | 0.5443 |
|  | SCS | 0.8255 | 0.8258 | 0.8247 | 0.8197 | **0.8271** | 0.8241 | 0.8228 |
|  | BPCCM | 0.5439 | 0.5421 | **0.561** | 0.5315 | 0.534 | 0.5359 | 0.5053 |
|  | SCTP | **0.898** | 0.8964 | **0.8977** | 0.877 | 0.8959 | **0.8981** | **0.8977** |
|  | HQC | 0.9678 | 0.9648 | **0.9688** | 0.965 | 0.9671 | 0.9671 | 0.9674 |
|  | IFD | 0.9132 | **0.9194** | 0.914 | 0.8794 | 0.912 | 0.8924 | 0.8864 |
|  | PSSDP | **0.2878** | 0.2824 | 0.2864 | 0.2735 | 0.2853 | **0.2878** | 0.281 |

Table 20: Performance of models trained with varying hyperparameter optimization regimes on private test competition datasets after standardized preprocessing. Higher values correspond to better performance. The original metric was reversed for SVPC, OGPCC, and BPCCM to align with the higher-is-better notation. The best model is highlighted. A model is considered better if it achieves a score that is one leaderboard standard deviation (std) larger than the other. The std is determined based on all top 1% submissions to only focus on the best models.

|  |  | XGBoost | LightGBM | CatBoost | ResNet | FTT | MLP-PLR | GRANDE |
|---|---|---|---|---|---|---|---|---|
| Default | MBGM | 0.5499 | 0.5473 | **0.5546** | 0.5485 | 0.5488 | 0.5524 | 0.5483 |
|  | SVPC | **0.4683** | 0.4643 | 0.4646 | 0.4369 | 0.457 | 0.4665 | 0.4409 |
|  | AEAC | 0.8947 | 0.8992 | **0.9039** | 0.8881 | 0.901 | 0.8168 | 0.816 |
|  | OGPCC | **0.5504** | 0.5473 | 0.5484 | 0.528 | 0.5327 | 0.5359 | 0.5227 |
|  | SCS | 0.8249 | **0.8261** | 0.8254 | 0.8203 | **0.8267** | 0.8239 | 0.8216 |
|  | BPCCM | 0.5672 | **0.5676** | 0.5668 | 0.5627 | 0.5667 | **0.5681** | 0.5651 |
|  | SCTP | 0.885 | 0.8939 | **0.8962** | 0.8643 | 0.8942 | 0.8906 | 0.8952 |
|  | HQC | 0.9618 | 0.9631 | **0.968** | 0.9661 | **0.968** | 0.9642 | 0.9628 |
|  | IFD | **0.9275** | 0.9255 | 0.9255 | 0.8759 | 0.9233 | 0.9173 | 0.9097 |
|  | PSSDP | 0.2841 | 0.283 | 0.2895 | 0.2848 | **0.2911** | 0.2779 | 0.2788 |
| Light | MBGM | 0.5534 | 0.5512 | **0.555** | 0.5503 | 0.551 | 0.5518 | 0.5531 |
|  | SVPC | **0.4694** | 0.4682 | 0.4642 | 0.4461 | 0.4582 | 0.4647 | 0.4653 |
|  | AEAC | 0.8941 | 0.9033 | **0.9059** | 0.9002 | 0.8981 | 0.8874 | 0.8619 |
|  | OGPCC | 0.5538 | **0.5589** | 0.5501 | 0.529 | 0.5403 | 0.5563 | 0.4857 |
|  | SCS | **0.8265** | 0.8261 | **0.8263** | 0.8212 | **0.8266** | 0.8248 | 0.8244 |
|  | BPCCM | 0.5666 | 0.5649 | 0.5678 | 0.5643 | 0.5664 | **0.5682** | **0.5688** |
|  | SCTP | 0.8961 | 0.8963 | **0.8973** | 0.868 | 0.8936 | **0.8973** | **0.8966** |
|  | HQC | **0.9686** | 0.9668 | **0.9686** | 0.9659 | **0.9684** | **0.9682** | 0.9673 |
|  | IFD | **0.9316** | 0.9289 | 0.9278 | 0.8684 | 0.922 | 0.9145 | 0.9237 |
|  | PSSDP | 0.2891 | 0.2889 | **0.2893** | 0.2887 | **0.2901** | **0.2894** | 0.2861 |
| Extensive | MBGM | 0.5522 | 0.5516 | **0.5537** | 0.55 | 0.5517 | 0.5521 | 0.5522 |
|  | SVPC | **0.4694** | 0.4682 | 0.4645 | 0.4446 | 0.4611 | 0.465 | 0.4626 |
|  | AEAC | 0.8972 | **0.9038** | 0.9028 | 0.8988 | 0.898 | 0.8892 | 0.8701 |
|  | OGPCC | **0.5615** | 0.56 | 0.557 | 0.5311 | 0.5415 | **0.5622** | 0.5444 |
|  | SCS | **0.8271** | **0.8271** | **0.8263** | 0.8209 | 0.8256 | 0.8252 | 0.8256 |
|  | BPCCM | 0.5659 | 0.5653 | **0.5692** | 0.5647 | 0.5662 | 0.5662 | **0.5689** |
|  | SCTP | 0.8982 | 0.8978 | 0.8987 | 0.8725 | 0.8942 | 0.8978 | **0.9002** |
|  | HQC | **0.9687** | 0.9667 | **0.9688** | 0.9663 | **0.9685** | 0.9684 | 0.9676 |
|  | IFD | **0.9319** | 0.9303 | 0.9268 | 0.8764 | 0.9196 | 0.9174 | 0.9229 |
|  | PSSDP | **0.2912** | 0.2902 | **0.2905** | 0.2883 | **0.2908** | **0.2909** | 0.2869 |

Table 21: Performance of models trained with varying hyperparameter optimization regimes on private test competition datasets after feature engineering. Higher values correspond to better performance. The original metric was reversed for SVPC, OGPCC, and BPCCM to align with the higher-is-better notation. The best model is highlighted. A model is considered better if it achieves a score that is one leaderboard standard deviation (std) larger than the other. The std is determined based on all top 1% submissions to only focus on the best models.

|         |       | XGBoost | LightGBM | CatBoost | ResNet | FTT | MLP-PLR | GRANDE |
|---------|-------|---------|----------|----------|--------|-----|---------|--------|
| Default | AEAC  | 0.9055 | 0.9066 | **0.9144** | 0.893 | 0.8967 | 0.8472 | 0.8388 |
|         | OGPCC | **0.5601** | 0.5578 | 0.5567 | 0.5312 | 0.5309 | 0.5452 | 0.5326 |
|         | SCS   | 0.8247 | 0.8251 | 0.8255 | 0.8214 | **0.8268** | 0.8253 | 0.823 |
|         | SCTP  | 0.9143 | 0.9154 | **0.9168** | 0.8797 | 0.9135 | 0.915 | **0.917** |
|         | IFD   | **0.9307** | **0.9301** | 0.9284 | 0.8755 | 0.9254 | 0.9129 | 0.9132 |
|         | PSSDP | 0.284 | 0.2834 | 0.2895 | 0.2865 | **0.2906** | 0.2741 | 0.2786 |
| Light   | AEAC  | 0.9069 | 0.9113 | **0.9171** | 0.9034 | 0.9052 | 0.898 | 0.8655 |
|         | OGPCC | 0.568 | **0.5729** | 0.5589 | 0.536 | 0.5412 | 0.5571 | 0.4878 |
|         | SCS   | **0.8271** | **0.8263** | **0.8262** | 0.8227 | **0.8269** | 0.8251 | 0.8242 |
|         | SCTP  | 0.9173 | 0.9172 | **0.9183** | 0.8864 | 0.9146 | **0.9193** | 0.9176 |
|         | IFD   | **0.9343** | 0.9327 | 0.9314 | 0.8741 | 0.9209 | 0.9183 | 0.9186 |
|         | PSSDP | **0.2901** | **0.2898** | **0.2898** | 0.2895 | **0.2905** | 0.2889 | 0.287 |
| Extensive | AEAC  | 0.9087 | 0.9115 | **0.9172** | 0.9019 | 0.9035 | 0.9032 | 0.872 |
|         | OGPCC | **0.5743** | **0.5748** | 0.5652 | 0.5333 | 0.5447 | 0.5638 | 0.551 |
|         | SCS   | **0.8273** | **0.8266** | **0.8269** | 0.82 | **0.8264** | 0.8259 | **0.8264** |
|         | SCTP  | **0.9193** | 0.9189 | **0.9194** | 0.8908 | 0.9144 | **0.9202** | **0.9196** |
|         | IFD   | **0.935** | **0.9352** | 0.9291 | 0.8698 | 0.9232 | 0.9186 | 0.9218 |
|         | PSSDP | **0.291** | **0.2906** | **0.2908** | 0.2894 | **0.2911** | **0.2914** | 0.2876 |

Table 22: Performance of models trained with varying hyperparameter optimization regimes on private test competition datasets after test-time adaptation. Higher values correspond to better performance. The original metric was reversed for SVPC, OGPCC, and BPCCM to align with the higher-is-better notation. The best model is highlighted. A model is considered better if it achieves a score that is one leaderboard standard deviation (std) larger than the other. The std is determined based on all top 1% submissions to only focus on the best models.

|      | MBGM | SVPC | AEAC | OGPCC | SCS | BPCCM | SCTP | HQC | IFD | PSSDP |
|------|------|------|------|-------|-----|-------|------|-----|-----|-------|
| Def. | 0.5505 | 0.366 | 0.8816 | **0.5979** | 0.827 | 0.565 | 0.8978 | **0.9689** | 0.871 | 0.2858 |
| FE   | **0.5524** | **0.454** | 0.9087 | 0.5873 | **0.8285** | **0.5678** | 0.8984 | **0.9692** | 0.902 | **0.2889** |
| TTA  | - | - | **0.9194** | 0.5971 | 0.8292 | - | 0.9194 | - | 0.908 | 0.2894 |

Table 23: Performance of AutoGluon on private test competition datasets after different preprocessing applied. Higher values correspond to better performance. The original metric was reversed for SVPC, OGPCC, and BPCCM to align with the higher-is-better notation. The best preprocessing is highlighted.

