# OpenReview forum: "A Data-Centric Perspective on Evaluating Machine Learning Models for Tabular Data"
_NeurIPS.cc/2024/Datasets_and_Benchmarks_Track — NeurIPS 2024 Track Datasets and Benchmarks Poster_

### Official Review · Reviewer_VLQP · 2024-07-23
**An extensive benchmark of tabular methods and tricks on real-world datasets**

**Rating:** 8
**Confidence:** 5
**Clarity:** The paper is very well written and cl…

**Review:**

The paper is very well written and extremely clear to follow. The experimental design is sound and the evaluation is extensive. The inclusion of multiple model families and an AutoML system leads to a well-rounded comparison. The usage of private leaderboard scores combined with provided code give a high degree of trustworthiness to the results. The paper covers an important topic of the impact of common data scientist tricks-of-the-trade in real-world tabular datasets, and provides a strong empirical baseline for future work to build upon.

While the evaluation is extensive, the provided code is not yet fully reproducible due to the lack of specified package versions ranges or a frozen requirements file. There is also no mention of the used versions of methods in the paper or appendix. 10 datasets, while a good start (especially given the human feature engineering steps), is still far from enough to have a strong empirical comparison of methods that is resilient to overfitting in follow-up studies. A top priority to increase the benchmark's impact will be to identify more competitions to include in the evaluation, either via Kaggle or other competition sites. The inclusion of more multiclass and regression tasks would be very useful. If more datasets were added, I foresee widespread adoption of the benchmark for future tabular method research. There is little to no discussion on the difference in training time / compute resources required for each method. A pareto frontier plot of leaderboard placement x training time (or cost to fairly compare CPU vs GPU) could potentially benefit the paper.


-----
REBUTTAL UPDATE
-----

I have raised my score from a 7 to an 8 after considering the author's detailed rebuttal response that expertly answered the vast majority of my questions and concerns. I strongly support the paper's acceptance and am willing to champion the paper.

**Strengths:**

Refer to the review above.

**Additional Feedback:**

The newest competition used in this paper is from 2019. What are the author's thoughts on how to identify and leverage more recent competitions? Perhaps a conversation with Kaggle staff could indicate if anything can be done for technical issues on availability / code competitions.

More of a thought than a limitation:
Meta Kaggle [1] provides the full timeline of submissions for each competition and the scores from each team. Using this, you can also calculate the position of each solution at a specific time in the competition. Kaggle competitions are prone to having more extreme solutions than in practical application, and the prevalence of the discussion forum, public leaderboard, and especially Kaggle Notebooks leads to the human scores by percentile being significantly stronger than what would occur without the extended sharing of information. A way to somewhat avoid this is by comparing with the first 24 hours or first 7 days of a competition, where there is less time for the sharing of knowledge and cloning of notebook solutions to impact the leaderboard. As an extension to the "Challenges" idea, for example "Being within top X% in the first 24 hours". This may also be a way to quantify the difficulty of a competition, by seeing how long it takes for human performance to saturate on a competition after it launches, or the amount the score deltas between leaderboard quantiles change over time (especially if they shrink and approach 0, such as from widespread notebook cloning and resubmission).

[1] https://www.kaggle.com/datasets/kaggle/meta-kaggle

**Correctness:**

The paper appears technically excellent in approach and execution. I have identified no flaws in the paper's evaluation, and the usage of the Kaggle private leaderboard leads me to have high trustworthiness in the reported numbers.

**Documentation:**

Yes, the datasets are public and the evaluation is conducted via submission to Kaggle. The authors should add package version details to maximize reproducibility.

**Ethics:**

I do not foresee any ethical concerns.

**Limitations:**

The authors have adequately addressed limitations.

**Opportunities For Improvement:**

## Major

There does not appear to be package versions listed in the paper or the code to reproduce the results. For example, the version of CatBoost, XGBoost, LightGBM, and especially AutoGluon are important to document. The code should come with a setup.py or requirements.txt, and ideally a requirements_frozen.txt for reproducibility. This is the most important point to address for me to consider increasing my score.

10 datasets, while a good start (especially given the human feature engineering steps), is still far from enough to have a strong empirical comparison of methods that is resilient to overfitting in follow-up studies. A top priority to increase the benchmark's impact will be to identify more competitions to include in the evaluation, either via Kaggle or other competition sites. The inclusion of more multiclass and regression tasks would be very useful. If more datasets were added, I foresee widespread adoption of the benchmark for future tabular method research.

## Minor

There is little to no discussion on the difference in training time / compute resources required for each method. A pareto frontier plot of leaderboard placement x training time (or cost to fairly compare CPU vs GPU) could potentially benefit the paper.

Given that AutoGluon fits many of the models baselined in this paper, it would be interesting to have a section discussing theories for why the base model outperforms AutoGluon on certain competitions (IFD, PSSDP, MBGM), even when sharing the same feature engineering steps. This would be valuable to the AutoML field as it highlights cases where the design decisions taken by the AutoML system may inhibit its ability to properly ingest high quality preprocessed data and vend it to the appropriate models. For example, maybe in cases where CatBoost outperforms AutoGluon, AutoGluon trains a very similar CatBoost model, but first preprocesses the data in a way that leads to worse results than the "standard" method (as defined in the paper), explaining the performance gap. One way to verify this is by fitting only a single CatBoost model in AutoGluon using the same hyperparameters as the baseline method, and seeing if AutoGluon is able to achieve similar results. If it is, then this would indicate that AutoGluon did not find the right hyperparameters of the model (a problem with model search space). If AutoGluon's version of the model achieves worse results, then it points to a problem in data preprocessing or data splitting in AutoGluon limiting the performance of its internal models.

Some models use GPU to train, whereas others use CPU. Is this a cause for major performance differences in the results? Are there cases where 10 hours was insufficient to properly train the model on CPU?

A nuance that would be good to mention is that the feature engineering tricks applied to a given competition are based on what was proven to work for the specific modeling technique those feature engineering tricks were used in. For example, a competition solution that used a tree model would likely have different feature engineering logic than one that used a neural network. This biases the performance of methods in this paper towards the method that originally was used in the competition solution.

The code has hardcoded method names in several locations. It might make sense for methods to have their own class definitions be used more frequently when determining this logic to reduce the number of hardcoding occurrences (ditto for datasets). The primary goal would be to simplify the effort required to add a new method / dataset to the benchmark without introducing unintended bugs or code duplication. (Example: L147, L194,  in `modeling_helpers.py`). An alternative could be: `if model_class.is_neural_net`, `if model_class.is_automl`, etc.

**Relation To Prior Work:**

The paper provides extensive citations to related work.

**Summary And Contributions:**

This paper benchmarks tree, neural network, and AutoML methods on 10 tabular datasets from Kaggle. It additionally compares the methods with HPO performed, manual feature engineering tricks, and test-time adaptation techniques. Running these comparisons, the authors highlight the private Kaggle leaderboard placement of the methods with the respective tricks enabled incrementally. They showcase that the gap in method performance decreases as more advanced techniques are applied, with feature engineering and test time adaptation being most important for top solutions. The authors find that 4 competitions have methods that achieve top 1% results without using dataset specific tricks (CatBoost and AutoGluon), whereas all 10 competitions have a method that achieves top 1% results after applying all tricks. They also provide code to reproduce the experiments and data preprocessing.

---

> ### Author Response · Authors · 2024-08-16
> **Answer to Reviewer Vlqp (1/3)**
>
> It was gratifying to read that the reviewer recognised the methodology and presentation of our paper and the potential impact of our contribution. We want to sincerely thank the reviewer for taking the time to examine our code and for making thoughtful suggestions for improvement.
>
>
> ### Reproducibility
> > There does not appear to be package versions listed in the paper or the code to reproduce the results. [...] The code should come with a setup.py or requirements.txt, and ideally a requirements_frozen.txt for reproducibility. This is the most important point to address for me to consider increasing my score.
>
> We apologize for the insufficient requirements description - thank you for pointing this out. We will add a requirements.txt and requirements_frozen.txt to our repository. For the rebuttal, please consider pip-installing the following package versions in a clean Python 3.11.7 environment from a .txt:
>
> ```
> autogluon==1.1.1
> catboost==1.2.5
> category-encoders==2.6.1
> GRANDE==0.1.6
> kaggle==1.6.14
> lightgbm==4.3.0
> matplotlib==3.9.0
> openfe==0.0.12
> optuna==3.6.1
> pytorch_frame==0.2.2
> rtdl_num_embeddings==0.0.9
> rtdl_revisiting_models==0.0.2
> setuptools==57.5.0
> tensorflow==2.16.1
> torch==2.2.1
> torchcontrib==0.0.2
> torcheval==0.0.7
> torchmetrics==1.2.1
> xfeat==0.1.1
> xgboost==2.0.3
> ```
>
>
> ### No. of datasets
> > 10 datasets, while a good start (especially given the human feature engineering steps), is still far from enough to have a strong empirical comparison of methods that is resilient to overfitting in follow-up studies.
>
> We agree that resilience to overfitting might be an issue with 10 datasets. However, this is less of an issue in our framework than it would be in conventional benchmarks. For details, see our response to Reviewer Wbvk.
>
> > A top priority to increase the benchmark's impact will be to identify more competitions to include in the evaluation
>
> We entirely agree that the top priority after publication is to gather more datasets, especially for regression and multi-class tasks. We believe that a larger benchmark of this kind can only evolve as a community effort. By releasing our framework as open-source code we do the first step.
>
> > If more datasets were added, I foresee widespread adoption of the benchmark for future tabular method research.
>
> We are honored that the high potential of our framework is recognized!
>
> ### Difference in training time / compute resources required for each method
>
> > There is little to no discussion on the difference in training time / compute resources required for each method. A pareto frontier plot of leaderboard placement x training time (or cost to fairly compare CPU vs GPU) could potentially benefit the paper.
>
> Thank you for pointing out that a discussion on training time could benefit our paper. We initially did not include such an analysis as we believe that other papers already do a great job in quantifying training time of most models in our scope (i.e., [2]).
>
> The pareto frontier plots for our experiments are pretty simple:
> * With standardized preprocessing, the only pareto optima are XGBoost and CatBoost.
> * With FE and FE-TTA, XGBoost is the only pareto optimum (fastest & best performing model at the same time) further supporting our claim that model selection is less important after expert-level preprocessing.
> * Only looking at deep models, MLP-PLR has the best leaderboard placement / training time ratio.
>
> Our results are in line with the observation that Kaggle participants often stick to fast-to-execute models like XGBoost and LightGBM. The plots will be added to the Appendix of our paper. For the rebuttal, we provide the table to produce the Pareto plots:
>
> |              |         |   ResNet |   GRANDE |   FTT |   MLP-PLR |   LightGBM |   XGBoost |   CatBoost |
> |:-------------|:--------|---------:|---------:|----------------:|----------:|-----------:|----------:|-----------:|
> | Standardized | Time    |   50.234 |  401.929 |         373.93  |    60.154 |     16.933 |    14.236 |    142.435 |
> |              | LB pos. |    0.444 |    0.519 |           0.594 |     0.603 |      0.622 |     0.677 |      0.786 |
> | FE           | Time    |   85.58  |  375.276 |         649.394 |    89.861 |     26.774 |    13.909 |    242.513 |
> |              | LB pos. |    0.623 |    0.808 |           0.809 |     0.821 |      0.849 |     0.901 |      0.885 |
> | FE-TTA       | Time    |   84.157 |  399.524 |         480.211 |    96.238 |     27.843 |    16.88  |    242.861 |
> |              | LB pos. |    0.641 |    0.825 |           0.906 |     0.907 |      0.902 |     0.975 |      0.97  |
>
> 'LB pos' denotes the leaderboard percentile after extensive HPO - the results are averaged over all datasets. 'Time' denotes the total time in minutes for training the model with default hyperparameters on all folds of all datasets.

---

> > ### Author Response · Authors · 2024-08-16
> > **Answer to Reviewer Vlqp (2/3)**
> >
> > ### Analysis of AutoGluon
> >
> > > Given that AutoGluon fits many of the models baselined in this paper, it would be interesting to have a section discussing theories for why the base model outperforms AutoGluon on certain competitions (IFD, PSSDP, MBGM), even when sharing the same feature engineering steps.
> >
> > Indeed, a detailed analysis of modes of failure for AutoGluon with preprocessed data would be interesting. Investigating that was out-of-scope for our paper, but we did a deeper investigation for the PSSDP dataset based on your suggestions.
> >
> > > maybe in cases where CatBoost outperforms AutoGluon, AutoGluon trains a very similar CatBoost model, but first preprocesses the data in a way that leads to worse results than the "standard" method (as defined in the paper), explaining the performance gap. One way to verify this is by fitting only a single CatBoost model in AutoGluon using the same hyperparameters as the baseline method, and seeing if AutoGluon is able to achieve similar results. [...] If AutoGluon's version of the model achieves worse results, then it points to a problem in data preprocessing or data splitting in AutoGluon limiting the performance of its internal models.
> >
> > We followed your suggestion and compared a CatBoost model with the same hyperparameters in AutoGluon vs. our implementation for dataset PSSDP after the test-time adaptation preprocessing pipeline. The results are as follows:
> >
> > |Model                   | LB percentile |
> > | ----------             | ----------    |
> > |CatBoost (ours)         | 68.5%         |
> > |CatBoost (AG)           | 51.4%         |
> >
> > The hyperparameters were:
> >
> > ``
> > 'iterations': 10000,
> > 'learning_rate': 0.055078095725390575,
> > 'random_seed': 0,
> > 'allow_writing_files': False,
> > 'eval_metric': 'Logloss',
> > 'depth': 4,
> > 'grow_policy': 'SymmetricTree',
> > 'l2_leaf_reg': 2.894432181094842,
> > 'max_ctr_complexity': 4,
> > 'one_hot_max_size': 10
> > ``
> >
> > Hence, there could indeed be a problem in data preprocessing or data splitting in AutoGluon. Unfortunately, making the AutoGluon preprocessing pipeline more accessible to users is an open issue on the official Github page since 2020 (Issue 570). We will add the debugging of AutoML failures as a possibility for future work.
> >
> > > Some models use GPU to train, whereas others use CPU. Is this a cause for major performance differences in the results? Are there cases where 10 hours was insufficient to properly train the model on CPU?
> >
> > We had the same assumption when first realizing that AutoGluon is underperforming - However increasing the training time to 48 hours did not lead to better results for the PSSDP dataset (Top 25.7% after 10 hours vs. top 29.1% after 48 hours).
> >
> >
> > ### Model-dependent feature engineering
> > > A nuance that would be good to mention is that the feature engineering tricks applied to a given competition are based on what was proven to work for the specific modeling technique those feature engineering tricks were used in.
> >
> > Thanks for pointing this out - we will add this nuance to the limitations. A general observation was that feature engineering techniques working well for one method often also work well for others. Only for one dataset, the expert solution (slightly) differentiated between tree-based and neural network preprocessing.
> >
> >
> > ### Code improvements
> >
> > To further increase the possibilities to contribute we will improve our implementation as suggested to simplify the effort required to add a new method / dataset:
> > * We changed our code to use class definitions of models instead of hard-coded functions at the lines in the code specified by the reviewer.
> > * We specified the minimal requirements to contribute new datasets and models.

---

> > > ### Author Response · Authors · 2024-08-16
> > > **Answer to Reviewer Vlqp (3/3)**
> > >
> > > ### Extension to more (recent) datasets
> > > > The newest competition used in this paper is from 2019. What are the author's thoughts on how to identify and leverage more recent competitions? Perhaps a conversation with Kaggle staff could indicate if anything can be done for technical issues on availability / code competitions.
> > >
> > > Besides solving technical issues, more (recent) competitions could be included by
> > > * Lowering the requirements on expert solution performance (achieving top 1% is quite hard - we could start lower)
> > > * Include datasets with temporal components requiring special preprocessing
> > > * Sacrifice usability and integrate challenges from other platforms.
> > > * Extend the competition search to non-featured competitions (without reward), likely at the cost of the quality of datasets and solutions.
> > > * Contact competition winners
> > >
> > > Most of these steps come with lowering quality requirements and some will become much easier as soon as the paper is published and available as a reference.
> > >
> > >
> > > ### Evaluation w.r.t. competition time
> > >
> > > > More of a thought than a limitation: Meta Kaggle [1] provides the full timeline of submissions for each competition and the scores from each team. Using this, you can also calculate the position of each solution at a specific time in the competition. [...] This may also be a way to quantify the difficulty of a competition.
> > >
> > > We are grateful for this valuable suggestion and implemented it as an additional feature of our evaluation framework. It is now possible to conduct our evaluation w.r.t. a specific point in time of the competition. I.e., the following table lists the leaderboard percentile of the best model after standardized preprocessing. The columns denote how many days passed after the start of the competition.
> > >
> > > |       |     1 |     7 |    14 |    21 |    28 |    45 |    60 |   end |
> > > |:------|------:|------:|------:|------:|------:|------:|------:|------:|
> > > | MBGM  | 0.997 | 0.999 | 0.998 | 0.997 | 0.998 | 0.996 | 0.996 | 0.996 |
> > > | SVPC  | 0.995 | 0.999 | 1     | 1     | 1     | 0.983 | 0.871 | 0.792 |
> > > | AEAC  | 0.98  | 0.997 | 0.995 | 0.99  | 0.988 | 0.981 | 0.958 | 0.953 |
> > > | OGPCC | 0.954 | 0.845 | 0.809 | 0.798 | 0.794 | 0.783 | 0.731 | 0.722 |
> > > | SCS   | 0.995 | 0.976 | 0.97  | 0.948 | 0.903 | 0.863 | 0.824 | 0.824 |
> > > | BPCCM | 0.579 | 0.525 | 0.473 | 0.463 | 0.44  | 0.413 | 0.407 | 0.374 |
> > > | SCTP  | 0.986 | 0.894 | 0.892 | 0.864 | 0.68  | 0.55  | 0.511 | 0.511 |
> > > | HQC   | 0.99  | 0.997 | 0.998 | 0.998 | 0.997 | 0.998 | 0.994 | 0.991 |
> > > | IFD   | 0.997 | 0.99  | 0.985 | 0.978 | 0.921 | 0.776 | 0.694 | 0.621 |
> > > | PSSDP | 0.989 | 0.967 | 0.888 | 0.844 | 0.78  | 0.696 | 0.638 | 0.638 |
> > >
> > > It can be seen that as the competition progresses, more sophisticated submissions enter the competition and the relative performance only using standardized preprocessing decreases.
> > >
> > > Note that this evaluation is biased as it assumes that the competitors actually used the submission that also scores best on the private leaderboard. Users can mitigate that by constructing the leaderboard with the (not often valid) assumption that competitors use the best public leaderboard solution.
> > >
> > > -----------
> > > ### References:
> > >
> > > [1] Grinsztajn, L., Oyallon, E., & Varoquaux, G. (2022). Why do tree-based models still outperform deep learning on typical tabular data?. Advances in neural information processing systems, 35, 507-520.
> > >
> > > [2] McElfresh, D., Khandagale, S., Valverde, J., Prasad C, V., Ramakrishnan, G., Goldblum, M., & White, C. (2024). When do neural nets outperform boosted trees on tabular data?. Advances in Neural Information Processing Systems, 36.
> > >
> > > [3] Roelofs, R., Shankar, V., Recht, B., Fridovich-Keil, S., Hardt, M., Miller, J., & Schmidt, L. (2019). A meta-analysis of overfitting in machine learning. Advances in Neural Information Processing Systems, 32.

---

> > > > ### Comment · Reviewer_VLQP · 2024-08-17
> > > > **Response to Author's Rebuttal**
> > > >
> > > > Thank you for the **excellent** rebuttal response. The authors have clearly answered all of my questions and concerns with precision. I have updated my score. I strongly support acceptance and will champion the paper.
> > > >
> > > > I believe the AutoML community in particular will follow this work with great interest, and hopefully will contact the authors prior to camera ready for additional input. I would suggest the authors make their code publicly available ASAP, as this is the best way to get adoption early on and give time for the AutoML community to provide valuable feedback for the camera ready (esp. AutoGluon developers). For reference, the [AutoML Conference 2024](https://2024.automl.cc/) is occurring between Sep 9th - 12th, and having the code available prior to that date would be ideal for spreading the word.
> > > >
> > > > ## Responses
> > > >
> > > > ### General
> > > >
> > > > For all comments I do not have a specific response to, consider the author's rebuttal answer to have fully addressed my concerns.
> > > >
> > > > ### CatBoost
> > > >
> > > > > We followed your suggestion and compared a CatBoost model with the same hyperparameters in AutoGluon vs. our implementation for dataset PSSDP.
> > > >
> > > > Very interesting. One possibility is the early stopping logic difference, as AutoGluon uses a custom designed adaptive early stopping logic rather than a fixed patience. The other is the difference in data splits. Looking at the code, it appears that AutoGluon could be fit in a different manner to more closely mimic the way other methods are fit, by fitting AutoGluon in the same logical code as the rest (the manual bagging logic which calls `run_fold` in `modeling_helpers.py`). With this approach, we would avoid fitting AutoGluon via `best_quality` and instead pass the validation data as `tuning_data` in the AutoGluon fit call to mimic the CatBoost training as closely as possible (eliminating the difference in the data splits). (note: It would be good to have a toggle for using ray vs sequential for ease of debugging, I had to do quite a bit of hacking to remove ray to get things working).
> > > >
> > > > I have tried the following edit to the AutoGluonModel fit code:
> > > >
> > > > ```python
> > > >     def fit(self,
> > > >             X_train, y_train,
> > > >             eval_set=None,
> > > >            ):
> > > >
> > > >         label = y_train.name
> > > >         data = pd.concat([X_train,y_train],axis=1)
> > > >
> > > >         if eval_set is not None:
> > > >             X_val = eval_set[0][0]
> > > >             y_val = eval_set[0][1]
> > > >             tuning_data = pd.concat([X_val, y_val], axis=1)
> > > >         else:
> > > >             tuning_data = None
> > > >
> > > >         self.model = TabularPredictor(label, eval_metric=self.params["eval_metric"],
> > > >                                       path=f"./logs/AutoGluon/{self.params['dataset_name']}_{self.params['exp_name']}",
> > > >                                      )
> > > >
> > > >         extra_params = ["hyperparameters", "verbosity", "num_stack_levels", "dynamic_stacking"]
> > > >         extra_kwargs = {k: self.params[k] for k in extra_params if k in self.params}
> > > >
> > > >         # FIXME: Hack to only fit CatBoost, TODO: Use same CatBoost hyperparameters and early stopping logic as non-AG
> > > >         extra_kwargs["hyperparameters"] = {"CAT": {"ag.stopping_metric": "roc_auc"}}
> > > >         # extra_kwargs["feature_generator"] = None  # TODO: This would skip AutoGluon's feature preprocessing, potentially changing results
> > > >
> > > >         self.model.fit(data,
> > > >                        tuning_data=tuning_data,
> > > >                        time_limit=self.params["time_limit"],
> > > >                        presets=self.params["presets"],
> > > >                        **extra_kwargs,
> > > >                       )
> > > >
> > > > ```
> > > >
> > > > There would still need to be more work done to completely match the non-AG CatBoost logic, but it is an interesting avenue. I may provide a further reply if I decide to dive deeper on this now that the authors have provided reproducible installation instructions.
> > > >
> > > > ### Extension to more (recent) datasets
> > > >
> > > > I completely understand the author's concerns with adding non-Kaggle competitions, and I think it is very reasonable to focus on Kaggle exclusively in the early stages. If the authors provide a extensible code-base for ease of incorporating other competition sources, maybe the community could help in adding additional sources.
> > > >
> > > > ### Evaluation w.r.t. competition time
> > > >
> > > > Excellent! I think this is very valuable. Maybe it could be good to incorporate this into some of the "Challenges", such as "AutoML system percentile in first 24 hours / 7 days", given that AutoML systems would ideally be a component of a user's toolkit when approaching a problem, and having a top performing solution early on would inform the user of promising paths to getting even better performance. "first 24 hours" would directly mimic the goal of the currently running [2024 Kaggle AutoML Grand Prix competition](https://www.kaggle.com/automl-grand-prix), which scores results after 24 hours, and where the current leading teams are largely authors of the major AutoML systems (1st: AutoGluon, 3rd & 5th: H2O Driverless AI, 4th: LightAutoML).

---

> > > > > ### Author Response · Authors · 2024-08-26
> > > > > **Answer to Reviewer Vlqp**
> > > > >
> > > > > # Answer VLQP 2
> > > > >
> > > > > Thank you once again for your valuable feedback and for your commitment to support the acceptance of our paper!
> > > > >
> > > > > We appreciate your positive assessment of the relevance of our work for the AutoML community. We believe there is great potential in optimizing preprocessing pipelines instead of models and ensembling. Hopefully, this paper and our publicly available code will be a starting point.
> > > > >
> > > > > Thank you for providing code adjustments to align our CatBoost training behavior with AutoGluon's. Finding the cause of AutoGluon models' low performance requires a deeper investigation of the AutoGluon pipeline. Therefore, we leave the debugging of the AutoGluon/AutoML error modes for future work and welcome any further helpful suggestions.
> > > > >
> > > > > > It would be good to have a toggle for using ray vs sequential for ease of debugging, I had to do quite a bit of hacking to remove ray to get things working
> > > > >
> > > > > We will add this as a possible configuration.
> > > > >
> > > > > > incorporate this into some of the "Challenges", such as "AutoML system percentile in first 24 hours / 7 days"
> > > > >
> > > > > We will add the additional challenge to the Appendix.
> > > > >
> > > > > -----
> > > > >
> > > > > We are always grateful for further suggestions and feedback!

---

### Official Review · Reviewer_wbvk · 2024-07-24
**Benchmark with interesting insights, but concerns for generalizability**

**Rating:** 8
**Confidence:** 4
**Clarity:** Yes.

**Review:**

The authors present a well-written paper with convincing arguments that the status-quo of standard preprocessing for 'normal' model evaluation is inadequate (they explicitly and, in my opinion, rightfully exclude AutoML from this consideration). It is unfortunate that the benchmarking suite only contains 10 datasets, of which only 2 regression and 1 multiclass classification. This is not a large enough benchmarking suite to really expose strengths and weaknesses of different methods systematically. Perhaps it is even small enough of a collection to overfit on it. However, because each dataset also includes a working expert solution (feature engineering) creating a larger benchmarking suite is very labor intensive and for that I would argue the choice is somewhat justifiable. Hopefully, this encourages others to contribute.

I have a problem with using the kaggle leaderboard as a metric, though (that is not already mentioned in app E). It often leads to suggestive wording like that avoid much of the nuance. In particular, solutions often get shared, and so the submission are far from independent (team) efforts. Second, it also includes a lot of chaff that either only did an example submission, constant predictor baseline, and some cases even score (much) worse than that. While it isn't inherently problematic when only comparing method directly to each other as is done in this paper, they do also create the risk for claims that are (unintentionally) deceiving (in my opinion). For example, this paper says "top 1% of all competition participants" without qualification, and the AutoGluon paper that this draws inspiration from claims "beating 99% of the participating data scientists" (who says all entries are by 'data scientists'? A genuine effort from a real data scientist will outperform a random forest baseline). It also obscure absolute differences (though this is mentioned in the appendix). I think more care should be given in the paper, even if only briefly, to better contextualize the metric.

A second problem with using Kaggle is that it relies on the evaluation on a single, hidden, test set. There is no repeated evaluations with different train/test splits, there isn't the possibility to assess the validity of the test data (indeed, the authors take care to level the playing field for SVPC by adding test data to the training set). It could hinder reproducibility in the future if a competition decides to withdraw their data (or worse, Kaggle ceases to exist).

I think the limitations currently in the appendix should very briefly be mentioned and referred to from the main paper, to make them more prominent. Kaggle competitions are a skewed version of reality at best, even if it's possibly the best we have publicly available. Other dataset repositories do also have realistic datasets available (though generally without expert solutions).

**Strengths:**

The authors clearly show the effect of expert feature engineering in the evaluation of models. They provide a reasonable benchmark that can contextualize scores with best human performance, something that's important to provide context to "state-of-the-art".

**Additional Feedback:**

I am a little confused on why AG is doing worse than CB on multiple problems, as AG should include well-tuned (though general) defaults for CB in its ensembles. Do you have an explanation for that?

**Correctness:**

Experiments are not repeated, no error bars are shown. Authors address this in appendix E. I would agree with them that it's unlikely that claims made in the paper are due to randomness in the evaluation procedure. Concerns for using Kaggle for evaluation and already mentioned in 'review'.

**Documentation:**

Provided code and documentation is clear enough. Code could be more reader-friendly (e.g., smaller functions, less nesting, type annotation), but at first glance seems documented and readable enough for modification.

**Limitations:**

The authors' section in the appendix is good (though see also 'review'). But I think the limitations should be more prominent in the main paper.

**Opportunities For Improvement:**

The benchmark could really use additional datasets, especially for tasks not binary classification. Without it, it's not clear that the benchmark can be used for those type of tasks.

**Relation To Prior Work:**

Yes.

**Summary And Contributions:**

The authors propose that for regular model evaluations, benchmarks should include dataset-specific processing.
They show in an evaluation on 10 datasets, that which such processing the model selection becomes a lot less relevant, and performance differences in general are much smaller. It also shows that typical evaluations with a standardized pipeline may not even get close to the best possible performance. Modern methods still benefit significantly from this manual feature engineering. They propose a benchmarking suite based on 10 tabular kaggle datasets, and provide for each of them a dataset-specific preprocessing pipeline based on top-scoring solutions for their respective competition. Finally, they address the fact that several/many datasets used as-if i.i.d. still have (weak) temporal relations which may be used to do test-time training. They propose to evaluate the methods based on the Kaggle leaderboard percentile, as a dataset-invariant metric.

---

> ### Author Rebuttal · Authors · 2024-08-16
>
> # Answer to Reviewer Wbvk
>
> We sincerely thank the reviewer for the thoughtful assessment of our paper. We are pleased to hear that the reviewer found our paper well-written and our arguments for a data-centric evaluation convincing.
>
> ### No. of datasets
> > only contains 10 datasets, of which only 2 regression and 1 multiclass classification. This is not a large enough benchmarking suite to really expose strengths and weaknesses of different methods systematically. Perhaps it is even small enough of a collection to overfit on it. However, because each dataset also includes a working expert solution (feature engineering) creating a larger benchmarking suite is very labor intensive and for that I would argue the choice is somewhat justifiable. Hopefully, this encourages others to contribute.
>
> We agree that resilience to overfitting might be an issue with 10 datasets. However, this is less of an issue in our framework than it would be in conventional benchmarks (e.g. [1, 2]), because:
>
> * The datasets in our framework are comparably large. I.e., the smallest dataset in our framework is still larger than 21/36 datasets in the TabZilla benchmark [2]. Overfitting large training datasets is much harder.
> * Roelofs et al. [3] found that at least 10,000 test examples is a reasonable minimum test set size to protect against adaptive overfitting in Kaggle challenges. All test sizes in our framework, except for the MBGM dataset, are at least ~50K. This is much larger than the test sizes of most datasets in conventional benchmarks, which are often less than 10K.
> * Test labels are unknown making it hard to purposefully overfit on particular samples
> * The need of submitting to Kaggle, although automated, is an additional overfitting barrier.
>
> We further appreciate the recognition of the high effort required to implement expert-level solutions. In this respect, our benchmark is the most extensive currently available.
>
> ### Kaggle leaderboard as a metric
>
> > It often leads to suggestive wording like that avoid much of the nuance. [...] For example, this paper says "top 1% of all competition participants" without qualification, and the AutoGluon paper that this draws inspiration from claims "beating 99% of the participating data scientists" (who says all entries are by 'data scientists'? .
>
> We share the reviewer’s concerns about the potential suggestiveness of the leaderboard as a metric. In our paper, we have consistently used factually accurate statements such as "top 1% of all competition participants" (e.g., lines 105, 292). To further clarify, we will add the following sentence after line 105: "Note that not all solutions on the leaderboard are expert-level, and leaderboard distributions can vary across datasets."
>
> > In particular, solutions often get shared, and so the submission are far from independent (team) efforts. Second, it also includes a lot of chaff [...] I think more care should be given in the paper, even if only briefly, to better contextualize the metric.
>
> We appreciate the reviewer's suggestion to better contextualize the metric. We will add a limitations section before reporting our results, where we will explicitly discuss leaderboard distributions and the suggestiveness of the metric.
>
> > A second problem with using Kaggle is that it relies on the evaluation on a single, hidden, test set. There is no repeated evaluations with different train/test splits
>
> The reviewer is correct that using a single train/test split limits our ability to assess the impact of randomness in the split itself. However, it is important to note that users of our framework can still evaluate the randomness of the training procedure—such as bagging CV splits, model-inherent randomness, and HPO—through repeated evaluations. Additionally, the risk of overfitting to specific data splits is mitigated by the large test set sizes employed in our framework.
>
> > there isn't the possibility to assess the validity of the test data
>
> We agree and want to emphasize that, regardless whether labels are provided or not, the validity of the test data depends on trusting the source. As most of our datasets are from leading companies and we have resolved issues such as data leakage, there is a low risk of invalid test data.
>
> > It could hinder reproducibility in the future if a competition decides to withdraw their data (or worse, Kaggle ceases to exist).
>
> Indeed, this could become an issue. However, considering Kaggle's enduring popularity and the long-standing availability of the datasets we included, we are confident that our framework will remain reproducible long enough to have a lasting impact.
>
>
> ### Why AutoGluon sometimes fails to outperform the single models it contains
>
> > I am a little confused on why AG is doing worse than CB on multiple problems
>
> Please refer to our response to Reviewer VLQP, where we conducted a small experiment to investigate this observation. It appears that certain steps in AutoGluon's preprocessing pipeline can lead to a decrease in performance when working with preprocessed data. Additionally, AutoGluon occasionally fails to correctly identify that high-cardinality ordinal features should be treated as categorical features, as observed in the AEAC dataset without expert-level preprocessing.
>
>
> ----------
> ### References
>
> [1] Grinsztajn, L., Oyallon, E., & Varoquaux, G. (2022). Why do tree-based models still outperform deep learning on typical tabular data?. Advances in neural information processing systems, 35, 507-520.
>
> [2] McElfresh, D., Khandagale, S., Valverde, J., Prasad C, V., Ramakrishnan, G., Goldblum, M., & White, C. (2024). When do neural nets outperform boosted trees on tabular data?. Advances in Neural Information Processing Systems, 36.
>
> [3] Roelofs, R., Shankar, V., Recht, B., Fridovich-Keil, S., Hardt, M., Miller, J., & Schmidt, L. (2019). A meta-analysis of overfitting in machine learning. Advances in Neural Information Processing Systems, 32.

---

> > ### Comment · Reviewer_wbvk · 2024-08-20
> > **Satisfied. Updated Scores.**
> >
> > Thank you very much for the well-written rebuttal. I think my concerns are adequately addressed, and I updated my scores accordingly. I still maintain that the generalisability may be limited on account of the number of datasets (especially for regression, multiclass classification), but I do not think this is something we can reasonably expect authors to change for this submission, and not something that should hinder publication.

---

### Official Review · Reviewer_Rzfd · 2024-07-24
**Review for "A Data-Centric Perspective on Evaluating Machine Learning Models for Tabular Data"**

**Rating:** 6
**Confidence:** 4

**Review:**

Quality. Overall, the paper is very well written and easy to comprehend. Furthermore, the illustrations, figures, and tables used to support the content are helpful and insightful. The methodology used, including tabular ML models and experiments, is also well-designed to study this topic. However, I have several concerns regarding the description of the tasks and the evaluation protocol (see comments below).

Clarity. The paper provides enough details to understand the conducted experiments at a high level. However, some details are missing (see comments below).

Originality / Significance. The paper addresses a very relevant and timely topic, confirming that dataset-specific preprocessing is a major factor for performance. While this conclusion is less surprising (but still interesting as an empirical study), the paper also provides many smaller insights into handling categoricals, the progress of DL, and test-time adaptation.

Pros
  * New datasets and evaluation protocols
  * Study on test-time adaptation

Cons
  * Focus solely on a single competition platform
  * Lack of clarity regarding terminology, experimental details and claims (see additional comments)
  * Lack of clear focus

The paper studies a relevant problem and provides insights into data-centric aspects of evaluating tabular ML. Unfortunately, while the paper aims to target many questions, it does not provide fully convincing answers and results and leaves many questions open. For these reasons and in its current status, I don't consider the paper ready for publication.

**Note:** The rebuttal promises to address open questions and clarify claims. With these changes, the paper will provide valuable insights and directions for future research in tabular ML. Thus, I raised my score from 4 to 6.

**Strengths:**

See my comments above (Review).

**Additional Feedback:**

[1] Clarification of the interpretation of the bar plots (i.e. Figure 3).
  * What does the length of the bar mean? For example, I would assume that since large parts of each bar are blue, the blue components have the largest impact on the overall performance, but the text says something else (line 190).
  * What does it mean if a colour is missing from a bar? For example, there is no green component for SVPC.

[2] Clarification of the evaluation protocol. Given the explanation, it is unclear how hyperparameter tuning, pre-processing, and feature engineering interact. For example, was HPO applied with or without "expert features"? If yes, was HPO repeated for the processed dataset?

[3] Reporting of ranking variance. I understand the whole procedure is computationally extensive. However, could you please comment on the variance across datasets, e.g., for Figure 4?

[4] Claim in Section 4.2. This analysis claims that "meaningful progress [..] in developing general purpose architectures" has been made (contrasting "custom-designed" networks that were used in the competition). While I would not disagree, this claim is very general and requires more discussion (which is likely out of the scope of this paper). Could you please elaborate on the differences between the two NN variants (I assume one of them is the mentioned "ResNet" baseline?) and how this represents the progress of the whole field?

[5] Claim in Section 5, line 346: "Furthermore, unlike previously claimed [34], categorical features can indeed be an important challenge for deep learning models". To my understanding, the related experiment for this is in lines 265-276, which does not consider deep learning models. Could you please clarify based on which results you came to this conclusion?

[6] Distinction between "model-specific pre-processing", "dataset-specific pre-processing" and "feature engineering" and recommendations for future work in lines 339ff. I would appreciate a definition of these terms so that I can understand the paper fully. This should also discuss domain-specific pre-processing (e.g., imputing missing values based on domain knowledge) and feature engineering (e.g., creating new columns). Could you please provide a definition of these steps in the context of your paper?

To provide some further context about why I am wondering about this, lines 339ff seem to imply that general-purpose models (and I think this includes AutoML solutions searching for the best-performing pipeline) ideally handle _all_ pre-processing steps, which seems to contradict lines 324ff, which say that data-centric preprocessing should be applied before comparing models. So, the exact implications/recommendations for future directions are unclear and a major concern from my side.

Minor comments:
  * [lines 305-312] I agree that the Covertype dataset might not be sufficient to assess modern ML approaches, and the data contains geospatial information. However, I do not fully understand this paper's exact discussion and claims about temporal components in this dataset. Yes, this dataset has been used in online learning settings, but the observed drift could likely stem from the heavy class imbalance that arises when "streaming" this data (I couldn't find any discussion on this in the related work). The paper introducing this dataset [11] introduced it as a standard tabular ML task without a temporal component. I believe there are better examples to make this point.
  * [lines 195-197] How was the Spearman coefficient computed on pipelines? I assume you measured the correlation between performance across datasets and not between pipelines?
  * [lines 86-90] While I agree that Kaggle can provide a useful source for challenging, relevant and interesting tasks, I am not fully convinced by the claim that Kaggle competitions "solve real-world problems". Also, many other competition platforms could be relevant for this type of work, e.g. https://mlcontests.com/state-of-competitive-machine-learning-2023/#platforms

**Clarity:**

Yes, overall, the paper seems well written; however, several critical details, claims, and terms need to be described better or clarified (see further feedback)

**Correctness:**

The paper contains several claims that I could not comprehend, and I'll list them in the "Further Feedback Section".

**Documentation:**

Since the results (to my understanding) depend on and are only valid wrt a snapshot of the leaderboard at a specific point in time, the exact date should be highlighted in the main text.

Please see my comments above and below.

**Ethics:**

No.

**Limitations:**

While I would prefer a short paragraph in the main paper, the appendix convincingly discusses this work's limitations.

**Opportunities For Improvement:**

More details on the choice and description of the datasets. I very much appreciate that the paper provides details about the selection procedure. However, it would be great to have more details about the chosen datasets, e.g. what task they cover, descriptive names and a link to the respective Kaggle competition in the main paper. Additionally, if the dataset is already part of prior dataset collections, a link to the OpenML ID (e.g. https://www.kaggle.com/competitions/amazon-employee-access-challenge/overview is probably related to https://www.openml.org/d/4135 and, thus, part of prior comparisons) would also be helpful better to understand these results in the context of prior work.

State the difference between pre-processing and feature engineering. The paper uses "preprocessing" and "feature engineering" heavily. From the context, it is not obvious whether these refer to the same step (independent of whether the respective step was derived from "expert" solutions).

More details on the choice and description of preprocessing. Collecting best practices for data engineering from Kaggle is a huge contribution and could lead to general improvements, which should be studied in detail. While this is not the main scope of the paper, it would be interesting to see whether any steps are task-agnostic and apply to other datasets as well.

**Relation To Prior Work:**

Yes, prior work is discussed well.

Minor comments: Consistency and details of the bibliography could be improved, e.g.
  * [25, 26] refer to the same paper
  * at least [1,2,3,5,6,19,20,36,38,46,50,62] do not contain publication venues and do contain author names like "I. C. i. J. L. L. M. P. H. A.". If these are references to the Kaggle competitions, this should be emphasized (or the entries should be rather a footnote than an entry in the bibliography)

**Summary And Contributions:**

The paper proposes a new evaluation framework for tabular ML, focusing on dataset-specific feature engineering steps. The authors select 10 Kaggle datasets and evaluate tabular ML with and without dataset-specific feature engineering steps of the respective winning solutions. They found that conducting this feature engineering generally improves performance and also reduces the performance difference across modelling solutions. Furthermore, they argue that correctly handling categorical features is important and that many tabular datasets contain temporal components that are often overlooked.

---

> ### Author Response · Authors · 2024-08-16
> **Answer to Reviewer Rzfd (1/4)**
>
> We sincerely thank the reviewer for the detailed and valuable feedback. We acknowledge that certain aspects could have been described more thoroughly and are confident that we will be able to address these concerns in the camera-ready version. We will use the additional page to enhance the task descriptions, provide more details on the evaluation protocol, and refine the formulation of our claims based on your suggestions. Please do not hesitate to let us know if any part of our response remains unclear.
>
> ### Details on the chosen datasets in the main paper
>
> > it would be great to have more details about the chosen datasets, e.g. what task they cover, descriptive names and a link to the respective Kaggle competition in the main paper. Additionally, if the dataset is already part of prior dataset collections, a link to the OpenML ID [...] would also be helpful better to understand these results in the context of prior work.
>
> We agree that adding more dataset details would be helpful. We've included links to the Kaggle challenges in the Name column of Table 1. We also added a column for task type and rounded dataset sizes to thousands to obtain space for this addition. For better contextualization with prior work, we add OpenML references (available for 6/10 datasets) to the table in Appendix A.1. However, the descriptive challenge names are rather long and remain in Appendix A.1 to avoid clutter in the main paper.
>
> ### Definitions of pre-processing and feature engineering
>
> > State the difference between pre-processing and feature engineering. [...] From the context, it is not obvious whether these refer to the same step [...]  Distinction between "model-specific pre-processing", "dataset-specific pre-processing" and "feature engineering" [...] Could you please provide a definition of these steps in the context of your paper?
>
> We agree that providing clear definitions of preprocessing and feature engineering (FE) in the context of our paper would enhance its clarity. With the additional space available in the camera-ready version, we will include these definitions at the beginning of Subsection 3.2.
>
> First, our paper takes a pipeline perspective. **Preprocessing** refers to a pipeline that combines a “set of techniques used prior to the application of a [model]” [1]. **Feature engineering** (FE) refers to techniques that “construct novel features from given data with the goal of improving predictive learning performance” [2]. Consequently, the two terms do not mean the same, but feature engineering is a subset of preprocessing.
>
> We investigate three preprocessing pipelines, two of which include feature engineering techniques. The standardized pipeline does not contain FE techniques and is **dataset-agnostic** because it is applicable to any dataset (details in Subsection 3.2). The feature engineering pipeline and the test-time adaptation pipeline are **dataset-specific** as they include feature engineering techniques selected to improve the performance on a particular dataset. (details in Appendix A.2).
>
> Additionally, some models require further **model-specific** preprocessing (e.g., standardizing regression targets for neural networks). This is applied after the investigated pipelines solely to be able to efficiently train a particular model. This particular preprocessing is not a part of our study, as we rely on established techniques that are generally recommended for any dataset (details in Appendix B.2).
>
> In addition to adding definitions, we will make sure that all formulations in our paper are consistent with these definitions. We will revise some phrases for clarity, such as in line 6: "real-world modeling pipelines often require dataset-specific preprocessing **which includes feature engineering**."" or line 70f: "models in practical applications typically follow dataset-specific preprocessing **pipelines containing feature engineering techniques**"
>
> > This should also discuss domain-specific pre-processing (e.g., imputing missing values based on domain knowledge) and feature engineering (e.g., creating new columns).
>
> For the datasets in our framework, the expert preprocessing pipelines primarily consist of feature engineering steps and do not include specialized domain-specific preprocessing techniques (see Appendix A.2). Domain-specific techniques that cannot be learned by models, even in theory, are incorporated during data loading to ensure meaningful comparisons. While some feature engineering techniques listed in Appendix A.2 are specific to individual datasets, they are more likely the result of thorough task-specific data exploration by data scientists rather than being guided by specific domain knowledge.

---

> > ### Author Response · Authors · 2024-08-16
> > **Answer to Reviewer Rzfd (2/4)**
> >
> > > To provide some further context about why I am wondering about this, **lines 339ff** seem to imply that general-purpose models (and I think this includes AutoML solutions searching for the best-performing pipeline) ideally handle all pre-processing steps, which seems to contradict **lines 324ff**, which say that data-centric preprocessing should be applied before comparing models. So, the exact implications/recommendations for future directions are unclear and a major concern from my side.”
> >
> > Thank you for pointing out that our wording may have caused confusion. The two highlighted passages are independent implications for different future directions:
> >
> > **324ff targets evaluation design**. We did not intend to give a general recommendation to always use data-centric setups.
> > The exact implications of this paragraph are that researchers should be aware a) whether their datasets are amenable to feature engineering, b) that standardized preprocessing setups treat models as AutoML systems, and, c) that true ceteris paribus (c.p.) comparisons are hard if some models (i.e., CatBoost) apply feature engineering internally and others don't.
> > We will revise our formulation to be more explicit and highlight that feature-engineered setups can be more suitable if a study aims for truly c.p. conditions, while standardized setups can be more suitable if models are expected to be capable of feature engineering.
> >
> > **339ff targets inspiration for new methods**. We did not intend to imply that general-purpose models should ideally handle all pre-processing steps - some are clearly too dataset-specific.
> > The exact implication of this paragraph is that researchers developing new models for tabular data could take inspiration from feature engineering techniques. Our study made it evident that there are transformations of the feature space which are not learned by models without manual feature engineering. As we focused on a pipeline perspective, a natural direction for future work would be to look at particular feature engineering techniques to uncover modes of failure of current models. We believe that the use of the term "automate" was inappropriate to make our point. To avoid false implications, we will: a) change the paragraph title to **Investigate why some feature engineering operations are not inherently learned by models** b) reformulate line 343f: "Future work could take a data-centric perspective and **take inspiration from feature engineering techniques when developing novel architecture components**."
> >
> > We would be happy if you could let us now whether the distinction and exact implications became clear.
> >
> > ### More details on the choice and description of preprocessing
> >
> > > More details on the choice and description of preprocessing. Collecting best practices for data engineering from Kaggle is a huge contribution and could lead to general improvements, which should be studied in detail. While this is not the main scope of the paper, it would be interesting to see whether any steps are task-agnostic and apply to other datasets as well
> >
> > We agree that collecting best practices for data engineering from Kaggle is a huge contribution. Although this is not the primary focus of our paper, we have gathered some valuable insights:
> >
> > * As we described in Appendix A.2, some feature engineering techniques apply across many tasks: groupby interactions of categorical and numeric features (4), two-order categorical interactions (3), feature selection (3), categorical frequency encoding (3), dimensionality reduction (2), three-order categorical interaction (2), 2-order arithmetic interactions (2), sum of missing values in a row (2), and sum of zeros in a row (2)
> > * The most frequently applied feature engineering steps include categorical features
> > * Feature interactions are frequently manually engineered while transformations of single features are rather rare
> > * Dimensionality reduction techniques (e.g. tSNE) are reoccurring, but mainly used for test-time adaptation
> >
> > We will add these insights to the main paper in Subsection 3.2
> >
> >
> >
> > ### Clarification of the interpretation of the bar plots (i.e. Figure 3).
> > >	What does the length of the bar mean?
> >
> > The y-axis scale of the bar plots in Figure 3 is the leaderboard position from worse (0) to best (1). As Reviewer VLQP summarized it, the plot should be interpreted as "the private Kaggle leaderboard placement of the methods with the respective tricks enabled incrementally". Therefore, the length of each bar indicates the performance gain relative to the previous configuration. We chose this visualization, as it is well suited to highlight our main claim, that for many datasets and models top performance is not reachable without feature engineering and/or test-time adaptation (compare max of blue/orange/green bars vs. max of red/purple bars). At the same time it visualizes that CatBoost sometimes achieves top performance even with default hyperparameters (blue bars).

---

> > > ### Author Response · Authors · 2024-08-16
> > > **Answer to Reviewer Rzfd (3/4)**
> > >
> > > Besides the incremental performance assessment through bar lengths, an additional comparison of bar heights (top of each bar) is possible, i.e., to assess model differences. We will adapt the legend to make the incremental interpretation more obvious.
> > >
> > > > What does it mean if a colour is missing from a bar? For example, there is no green component for SVPC.
> > >
> > > A missing color means no performance gain over the preceding setting.
> > >
> > >
> > >
> > > ### Claim on progress of general-purpose architectures in Section 4.2.
> > >
> > > > This analysis claims that "meaningful progress [..] in developing general purpose architectures" has been made (contrasting "custom-designed" networks that were used in the competition). While I would not disagree, this claim is very general and requires more discussion (which is likely out of the scope of this paper). Could you please elaborate on the differences between the two NN variants (I assume one of them is the mentioned "ResNet" baseline?) and how this represents the progress of the whole field?
> > >
> > > The tabular ResNet architecture was introduced by Gorishny et al. [3] as a baseline for tabular DL and has been frequently used for that purpose in related work. Because it essentially is a MLP with residual connections and batch normalization (both introduced in 2015) it is representative of a generic tabular architecture before the increased interest in general-purpose deep learning architectures for tabular data.
> > >
> > > Our analysis reveals two insights:
> > >
> > > * The gap between ResNet and the best of the tree more recent architectures is substantial (Figure 5).
> > > * In two competitions, (custom) neural networks were originally the single-best models (SCTP, PSSDP - see Table 1). In our analysis, MLP-PLR and FTTransformer are able to reach top ranks on these datasets after feature engineering and test-time adaptation, while ResNet performs worse (see Figure 3).
> > >
> > > Especially the latter indicates that modern general-purpose architectures reduce the necessity of custom-designed networks. We acknowledge that we did not present these insights clearly enough. We will fix that for the camera-ready version. For additionally improved clarity we will:
> > > * Reference Table 1 and Figure 3 for making the relation to custom-designed networks understandable
> > > * Adapt the legend of Figure 5: "NN Baseline" --> "ResNet baseline"
> > > * Add "Best NN denotes the best model of FTTransformer, MLP-PLR, and GRANDE" to the caption of Figure 5.
> > >
> > > ### Claim on categorical features in Section 5, line 346
> > >
> > > > "Furthermore, unlike previously claimed [34], categorical features can indeed be an important challenge for deep learning models". To my understanding, the related experiment for this is in lines 265-276, which does not consider deep learning models. Could you please clarify based on which results you came to this conclusion?
> > >
> > > You are correct that lines 265-276 focus solely on two tree-based models. The conclusion referenced in line 346 is actually derived from the observation that deep learning models significantly improve their performance on categorical data when feature engineering techniques are applied. For instance, in the AEAC dataset, which consists entirely of categorical features, all neural networks perform poorly without feature engineering (e.g., categorical feature interactions), as shown in Figure 3. This suggests that current architectures do not adequately capture the complex patterns within categorical data.
> > >
> > > We realize that these observations need to be explicitly described in the experiments to better support the claim in line 346ff. We will include this explanation in Subsection 4.3 and extend line 346ff to incorporate tree-based models as a natural conclusion to the discussion in lines 265-276.
> > >
> > > ### Minor clarifications and improvements
> > >
> > > > “Since the results (to my understanding) depend on and are only valid wrt a snapshot of the leaderboard at a specific point in time, the exact date should be highlighted in the main text.“
> > >
> > > The snapshot is always the end of the competition. We will add this information to the Evaluation paragraph in Subsection 3.3. Additionally, we will provide the exact dates in the Appendix, as we believe that the year mentioned in Table 1 is sufficient detail for the main paper.
> > >
> > > > Clarification of the evaluation protocol. Given the explanation, it is unclear how hyperparameter tuning, pre-processing, and feature engineering interact. For example, was HPO applied with or without "expert features"? If yes, was HPO repeated for the processed dataset?
> > >
> > > For each model-dataset-preprocessing combination, we trained with three HPO settings (default, light, extensive). Detailed results of all settings are in the Appendix. Thanks to your feedback, we realized that only Figures 3 and 6 explicitly mention which HPO settings are exactly compared. We will add a sentence at the beginning of Section 4 that performance is reported after extensive HPO whenever not stated otherwise.

---

> > > > ### Author Response · Authors · 2024-08-16
> > > > **Answer to Reviewer Rzfd (4/4)**
> > > >
> > > > > Reporting of ranking variance. I understand the whole procedure is computationally extensive. However, could you please comment on the variance across datasets, e.g., for Figure 4?
> > > >
> > > > Please find the average LB positions and standard deviations over datasets in the following table:
> > > >
> > > > |        | CatBoost    | XGBoost     | LightGBM    | MLP-PLR     | FTTransformer   | GRANDE      | ResNet      |
> > > > |:-------|:------------|:------------|:------------|:------------|:----------------|:------------|:------------|
> > > > | Stand. | 0.79 (0.21) | 0.68 (0.17) | 0.62 (0.19) | 0.6 (0.22)  | 0.59 (0.19)     | 0.52 (0.23) | 0.44 (0.21) |
> > > > | FE     | 0.89 (0.14) | 0.9 (0.15)  | 0.85 (0.19) | 0.82 (0.19) | 0.81 (0.2)      | 0.81 (0.17) | 0.62 (0.27) |
> > > > | TTA    | 0.97 (0.04) | 0.97 (0.03) | 0.9 (0.16)  | 0.91 (0.12) | 0.91 (0.1)      | 0.83 (0.16) | 0.64 (0.28) |
> > > >
> > > > In general, the dataset-variance is rather high, mainly because the hardness of the task differs among datasets, especially with standardized preprocessing. It can be seen that the variance of the best performing methods strongly decreases after TTA, as good preprocessing eases the prediction tasks for the previously hard datasets.
> > > >
> > > >
> > > > > I do not fully understand this paper's exact discussion and claims about temporal components in [the covertype] dataset. [...] I believe there are better examples to make this point.
> > > >
> > > > We agree that better examples exist, and will instead use the electricity dataset (openml-id: 151). It is used in recent tabular data benchmarks [4,5] as well as for time-related concept drift [6]. Hence, we can replace the reference dataset without changing the claims.
> > > >
> > > > > How was the Spearman coefficient computed on pipelines? I assume you measured the correlation between performance across datasets and not between pipelines?
> > > >
> > > > For each preprocessing pipeline, we constructed a vector of all performances after extensive HPO (10 datasets x 7 models) and calculated the spearman coefficient of two pipeline vectors.
> > > >
> > > >
> > > > > Focus solely on a single competition platform
> > > >
> > > > Unfortunately, including more platforms was not possible with the current infrastructure of existing platforms. Kaggle, with its large user base, stands out as the platform where top solutions are almost always shared, competition datasets remain accessible online, and post-competition submissions are possible. Additionally, Kaggle provides an API that for the easy automation of submissions, which is crucial for the user-friendliness of our framework. In comparison, other platforms do not offer the same capabilities—Codalab, for example, has disabled new submissions to old challenges, and Zindi lacks an API for automating submissions. Consequently, we decided to rely solely on Kaggle. We hope that other platforms will evolve in the future, enabling us to include more datasets to our framework.
> > > >
> > > > ------------------------
> > > > Once again, thank you for highlighting these important aspects. Your feedback has been invaluable in helping us improve our paper!
> > > >
> > > > ------------------------
> > > >
> > > >
> > > > [1] García, S., Ramírez-Gallego, S., Luengo, J., Benítez, J. M., & Herrera, F. (2016). Big data preprocessing: methods and prospects. Big data analytics, 1, 1-22.
> > > >
> > > > [2] Khurana, U., Turaga, D., Samulowitz, H., & Parthasrathy, S. (2016, December). Cognito: Automated feature engineering for supervised learning. In 2016 IEEE 16th international conference on data mining workshops (ICDMW) (pp. 1304-1307). IEEE.
> > > >
> > > > [3] Gorishniy, Y., Rubachev, I., Khrulkov, V., & Babenko, A. (2021). Revisiting deep learning models for tabular data. Advances in Neural Information Processing Systems, 34, 18932-18943.
> > > >
> > > > [4] Grinsztajn, L., Oyallon, E., & Varoquaux, G. (2022). Why do tree-based models still outperform deep learning on typical tabular data?. Advances in neural information processing systems, 35, 507-520.
> > > >
> > > > [5] McElfresh, D., Khandagale, S., Valverde, J., Prasad C, V., Ramakrishnan, G., Goldblum, M., & White, C. (2024). When do neural nets outperform boosted trees on tabular data?. Advances in Neural Information Processing Systems, 36.
> > > >
> > > > [6] de Barros, R. S. M., de Carvalho Santos, S. G. T., & Júnior, P. M. G. (2016, July). A boosting-like online learning ensemble. In 2016 international joint conference on neural networks (IJCNN) (pp. 1871-1878). IEEE.

---

> > ### Comment · Reviewer_Rzfd · 2024-08-26
> > **Thanks a lot for your time and effort; score raised to 6**
> >
> > Overall, I am happy with this extensive rebuttal. Thank you very much, and I look forward to seeing the revised version.
> >
> > ----
> >
> > Minor comments
> >
> > > We would be happy if you could let us now whether the distinction and exact implications became clear.
> >
> > Yes, thanks a lot. This addresses my concern regarding the implications and recommendations for future work.
> >
> > > A missing color means no performance gain over the preceding setting.
> >
> > Ah, okay, thanks for the clarification.
> > [as a minor recommendation] I assume there is no guarantee that the settings always perform better or equal, right? So, this information would be lost if a setting decreases performance, e.g., due to overfitting. Maybe you want to consider a different visualization that additionally shows this aspect of the results.
> >
> > > For instance, in the AEAC dataset, which consists entirely of categorical features, all neural networks perform poorly without feature engineering (e.g., categorical feature interactions), as shown in Figure 3. This suggests that current architectures do not adequately capture the complex patterns within categorical data.
> >
> > Yes, I would agree with this conclusion. Thanks a lot for rephrasing.

---

### Author Response · Authors · 2024-08-16
**Rebuttal Summary**

# Rebuttal Summary

We thank the reviewers for taking the time to read our paper and provide valuable and detailed feedback.

In this answer we
1. Summarize positive aspectes highlighted by the reviewers
2. Summarize changes to our manuscript and how we use the additional page in the camera-ready version

Reviewer comments are adressed in the individual replies.

## Positive aspects
* **Clarity and Presentation** `[Rzfd, Wbvk, Vlqp]`: "well-written paper with convincing arguments" `[Wbvk]`; "the illustrations, figures, and tables used to support the content are helpful and insightful" `[Rzfd]`
* **Methodology and Evaluation** `[Rzfd, Wbvk, Vlqp]`: "methodology used, including tabular ML models and experiments, is also well-designed to study this topic" `[Rzfd]`; "experimental design is sound and the evaluation is extensive [...] high degree of trustworthiness to the results" `[VLQP]`
* **Novelty and significance** `[Rzfd, Vlqp]`: "addresses a very relevant and timely topic" `[Rzfd]`; "important topic of the impact of common data scientist tricks-of-the-trade in real-world tabular datasets" `[VLQP]`
* **Empirical usefulness** `[Wbvk, Vlqp]`: "strong empirical baseline for future work to build upon. [...] If more datasets were added, I foresee widespread adoption of the benchmark for future tabular method research." `[VLQP]`; "Provided code and documentation is clear enough [...] documented and readable enough for modification." `[Wbvk]`

## Changes to our Manuscript

### Minor changes to submitted manuscript and code
* `[Rzfd]`: Corrected error in formatting of dataset references
* `[Rzfd]` Changed Table 1 to include competition links and task types
* `[Rzfd]`: Minor changes to better distinguish feature engineering and preprocessing throughout the paper
* `[Rzfd]`: Reformulated implications for future work in 324ff and 339ff to be more explicit and clearly distinguished
* `[Rzfd]`: Improved information on how to interpret Figure 3
* `[Rzfd]`: Reference another dataset in line 308 to make our point on tabular data with temporal characteristics
* `[Wbvk]`: Add a note to raise awareness about the suggestiveness of using the leaderboard as a metric.
* `[Wbvk, VLQP]`: Added analysis of AutoGluon modes of failure to implications for future work.
* `[VLQP]`: Added a requirements file to our code
* `[VLQP]`: Removed hard-coded model behavior at two points in the code to ease further contributions



### How we use the additional page
* `[Rzfd, Wbvk]`: Added a short paragraph on limitations to the main paper (previously in Appendix)
* `[Rzfd]`: Added definitions on preprocessing and feature engineering to Subsection 3.2
* `[Rzfd]`: Add a paragraph on our insights about task-agnostic feature engineering techniques to Subsection 3.2
* `Rzfd]`: Added a more detailed interpretation of our results in Section 4.2. to present our claim on the progress of general-purpose architectures more clearly
* `[Rzfd]`: Added interpretation on our results regarding deep learning on categorical data to Subsection 4.3 to make the implications in Section 5, line 346 better understandable

### Additions to the Appendix
* `[Rzfd]`: Added openml ids and competition end dates to the Table in Appendix A.1
* `[Rzfd]`: Added a Table with model ranking variances across datasets
* `[VLQP]`: Added information on model runtimes in our framework
* `[VLQP]`: Added evaluation at certain points in time of the competitions (e.g., first 24 hours)

---

### Decision · Program_Chairs · 2024-09-26

**Decision:**

Accept (Poster)

**Comment:**

The paper executes a benchmark on various preprocessing operators, and their effect on algorithm performance. The study seems novel and executed correctly, filling a clear gap in our community. Additionally, all reviewers seem to be in agreement that this warrants publication at NeurIPS. For this reason, I would recommend acceptance.

However, I do want to clearly point out to the authors that the current version that I can inspect does not take into consideration all suggestions made by the reviewer: Reviewer Rzfd pointed out that the reference section is not in good order. I have checked the current state of the references, and this has not been updated yet. We participate in a system where credit assignment is important, and by submitting your paper for review, you are implicitly asking your peers to spend a considerate amount of time on the paper. If this is not met by a similar rigor on the side of the authors, the paper was not yet ready for submission. Because of the overwhelmingly positive reviews, I will ignore this, but I would impress upon the authors to fix this before the camera ready copy deadline.